# Affinity capture of polyribosomes followed by RNAseq (ACAPseq), a discovery platform for protein-protein interactions

Xi Peng[1,2†], Francesco Emiliani[1†], Philip M Smallwood[1,3], Amir Rattner[1], Hong Lei[1], Mark F Sabbagh[1,4], Jeremy Nathans[1,3,4,5]*

[1]Department of Molecular Biology and Genetics, Johns Hopkins University School of Medicine, Baltimore, Maryland, United States; [2]Department of Ophthalmology, West China Hospital, Sichuan University, Chengdu, China; [3]Howard Hughes Medical Institute, Johns Hopkins University School of Medicine, Baltimore, Maryland, United States; [4]Department of Neuroscience, Johns Hopkins University School of Medicine, Baltimore, Maryland, United States; [5]Department of Ophthalmology, Johns Hopkins University School of Medicine, Baltimore, Maryland, United States

**Abstract** Defining protein-protein interactions (PPIs) is central to the biological sciences. Here, we present a novel platform - Affinity Capture of Polyribosomes followed by RNA sequencing (ACAPseq) - for identifying PPIs. ACAPseq harnesses the power of massively parallel RNA sequencing (RNAseq) to quantify the enrichment of polyribosomes based on the affinity of their associated nascent polypeptides for an immobilized protein 'bait'. This method was developed and tested using neonatal mouse brain polyribosomes and a variety of extracellular domains as baits. Of 92 baits tested, 25 identified one or more binding partners that appear to be biologically relevant; additional candidate partners remain to be validated. ACAPseq can detect binding to targets that are present at less than 1 part in 100,000 in the starting polyribosome preparation. One of the observed PPIs was analyzed in detail, revealing the mode of homophilic binding for Protocadherin-9 (PCDH9), a non-clustered Protocadherin family member.
DOI: https://doi.org/10.7554/eLife.40982.001

*For correspondence:
jnathans@jhmi.edu

†These authors contributed equally to this work

## Introduction

Current platforms for large scale unbiased identification of PPIs utilize a variety of methods, each of which has strengths and limitations (*Gonzalez, 2012*; *Legrain and Rain, 2014*). Mass spectrometry of purified protein complexes provides an unbiased approach to identifying PPIs, but it is limited to relatively high affinity interactions as well as by sample quantity and purity (*Gavin et al., 2011*; *Frei et al., 2012*). Yeast and mammalian two-hybrid systems use libraries that direct fusion protein expression intracellularly and are therefore not readily applicable to extracellular domains (ECDs), which generally do not fold correctly in the cytoplasm (*Fiebitz et al., 2008*; *Rajagopala, 2015*). Direct binding assays using immobilized fusion proteins have high sensitivity but require production, purification, and analysis of hundreds or thousands of proteins (*Bushell et al., 2008*; *Ramani et al., 2012*; *Özkan et al., 2013*; *Tom et al., 2015*; *Visser et al., 2015*; *Hsu et al., 2017*). Identifying PPIs based on a bioassay has high specificity and biological relevance but requires some knowledge of protein function (e.g. *Lin et al., 2008*; *Zhang et al., 2014*). Proximity labeling assays, such as BioID, tag nearby proteins but only a subset of these is likely to interact directly with the protein of interest (*Fernández-Suárez et al., 2008*; *Roux et al., 2012*; *Schopp et al., 2017*).

We have developed an approach to identifying PPIs in which immobilized protein baits are used to capture polyribosomes via their binding to ribosome-associated nascent polypeptide targets. The identities of the targets are revealed by NextGen sequencing of the captured mRNA. This method was inspired by the work of Schimke and colleagues, who developed immuno-affinity capture of polyribosomes as a method for purifying specific eukaryotic mRNAs (*Palmiter et al., 1972*). Immuno-affinity capture of polyribosomes was subsequently used to enrich several low abundance mRNAs to facilitate their cloning (*Korman et al., 1982*; *Kraus and Rosenberg, 1982*). More recently, affinity capture of polyribosomes followed by ribosome profiling has been used to interrogate interactions between nascent polypeptides and chaperones or targeting factors, such as signal recognition particle (*Schibich et al., 2016*; *Döring et al., 2017*). A further variation on this method, ribosome display, was developed as an in vitro approach for identifying high affinity interactions between libraries of mutant proteins and an immobilized ligand (*Zahnd et al., 2007*; *Plückthun, 2012*; *Schilling et al., 2014*). The ribosome display technique has been further adapted by *Zhu et al., 2013* for the discovery of PPIs and protein-drug interactions by in vitro transcription, translation, and ribosome capture starting from a normalized collection of human cDNAs, a method referred to as parallel analysis of translated open reading frames (PLATO). ACAPseq is conceptually similar to PLATO.

In developing ACAPseq, we focused on mammalian protein domains that reside in the extracellular space, because (1) many of these proteins have biologically important interactions, (2) many extracellular binding interactions involve domains that are likely to fold independently (e.g. cadherin, fibronectin, and Ig domains), (3) most single-pass transmembrane proteins, including many cell surface receptors, consist of an amino-terminal extracellular domain (ECD) and a carboxy-terminal intracellular domain (ICD), the optimal arrangement of the ECD for a method that relies on binding in the context of a nascent polypeptide, and (4) most extracellular domains contain one or more disulfide bonds and, in general, these domains do not fold properly in the reducing environment of the cytoplasm and are therefore not amenable to yeast or mammalian two-hybrid approaches. Moreover, the potential for poly-valent interactions between several immobilized bait proteins and several nascent polypeptides associated with a single polyribosome suggested that ACAPseq might be useful for identifying relatively weak interactions.

## Results

### ACAPseq methodology

Protein baits were produced as secreted human IgG1 Fc fusion proteins following transient transfection of mammalian (HEK/293T) cells, immobilized on the surface of Protein-G-coated magnetic beads, and used to capture polyribosomes prepared from neonatal mouse brain, a tissue that expresses a large and diverse set of genes (*Figure 1A*). Capturing polyribosomes with multiple baits in parallel permitted a comparison of capture efficiencies among different baits, minimized batch effects in library preparation and NextGen sequencing, and was cost-effective since multiple barcoded libraries could be sequenced simultaneously. Typical RNA yields from ACAPseq capture reactions were in the several nanogram range. In our experience,~10 million RNAseq reads per bait are sufficient for identification of captured mRNAs.

*Figure 1B* shows genome browser images of exonic RNAseq reads from polyribosomes captured by the extracellular domains of EphrinA1 (EFNA1) and Fibronectin leucine-rich repeat transmembrane protein-3 (FLRT3), displayed as Fc fusions, together with RNAseq reads from the starting neonatal mouse brain polyribosome preparation. [In this and all other genome browser images, each panel shows the aligned RNAseq read coverage from parallel ACAPseq reactions that started with equal amounts of a common polyribosome stock solution, unless stated otherwise.] EFNA1-Fc specifically captured Erythropoietin-producing hepatocellular receptor a5 (Epha5) mRNA, and FLRT3-Fc specifically captured Adhesion G protein-coupled receptor L3 (Adgrl3) mRNA, both of which reside on mouse chromosome 5. The high signal-to-noise ratio for this pair of ACAPseq bait-target combinations is apparent in the genome browser image spanning the entire length of chromosome 5 (*Figure 1B*, upper panel) and likely reflects the low level of non-specific polyribosome trapping by the non-porous magnetic beads. This result serves as a technical validation, as both interactions have been described previously (*Frisén et al., 1999*; *Himanen et al., 2009*; *Lu et al., 2015*; *Ranaivoson et al., 2015*). Based on RNAseq read counts, Epha5 and Adgrl3 transcripts were

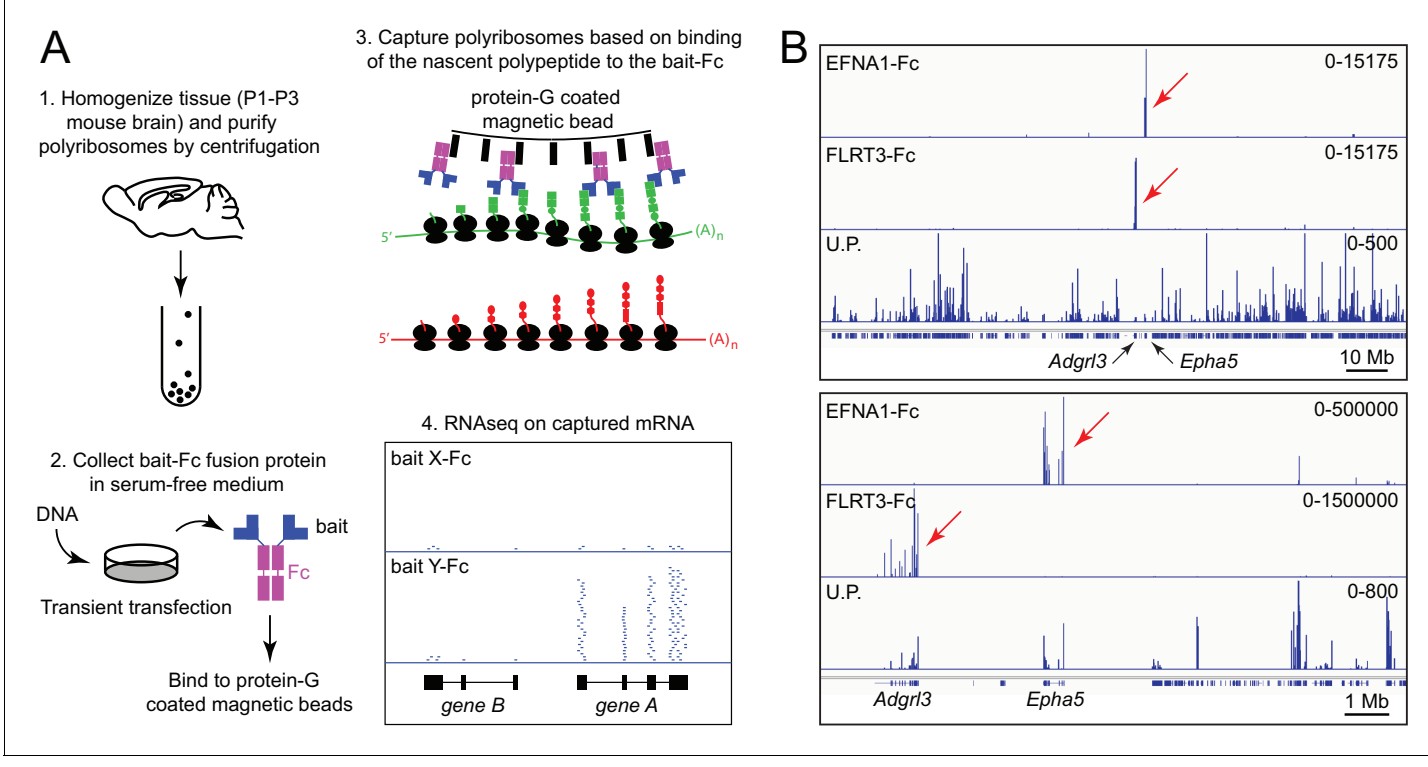

**Figure 1.** Principle of ACAPseq. (**A**) Workflow for the ACAPseq protocol. In step 3, the polyribosome synthesizing the green polypeptide is captured by magnetic beads displaying the bait-Fc fusion protein, and the polyribosome synthesizing the red polypeptide is not captured. In step 4, the genome browser image shows, at the bottom, introns (thin back lines) and exons (filled black rectangles) for genes A and B. The Y-Fc bait has captured polyribosomes synthesizing protein A, resulting in a large number of RNAseq reads (short horizontal blue lines) that align to the exons of gene A. (**B**) Histogram of mapped ACAPseq reads using EFNA1-Fc and FLRT3-Fc baits to capture neonatal mouse brain polyribosomes. The bottom track shows the unselected polyribosome sample. The upper panel spans the entirety of mouse chromosome 5 (151 Mb), and the lower panel spans a 10 Mb region encompassing the *Adgrl3* and *Epha5* genes. Red arrows highlight reads mapping to *Adgrl3* and *Epha5*. In this and all other genome browser images, the number range at the right side of each panel indicates the number of reads corresponding to the vertical height of the panel. U.P., unselected polyribosomes.

DOI: https://doi.org/10.7554/eLife.40982.002

present in the whole brain polyribosome preparation at abundances of 0.014% and 0.080%, and were each enriched ~500 fold by bait capture.

In the paragraphs that follow, we describe (1) the quality and reproducibility of the primary ACAPseq data using a diverse set of baits, (2) quantitative analyses of ACAPseq performance using genome-wide visualization of capture efficiency and with different thresholds for fold enrichment of target RNAs, (3) validation of a subset of bait-target combinations, and (4) an in-depth investigation of the binding interactions responsible for a novel PPI.

## ACAPseq with diverse mammalian ECD baits

Varying the level of either the EFNA1-Fc bait or the polyribosomes over a 25-fold range [0.4, 2, or 10 ml of serum-free conditioned medium (SFCM) containing EFNA1-Fc; and 4, 20, or 100 ug polyribosomes] produced little variation in capture efficiency for multiple bona fide targets, in the present instance Epha3-Epha8 mRNAs (*Figure 2A* and *Figure 2—figure supplement 1A–E*). The relative insensitivity of ACAPseq to bait levels implies that EFNA1-Fc saturates the magnetic beads at each of these three levels of serum-free conditioned medium. The relative insensitivity of ACAPseq to polyribosome dilution suggests that the affinities of EFNA1 for nascent EPHA receptors is sufficiently high that the polyribosome concentrations used here are also saturating. Another EFNA1-Fc target, HECT domain E3 ubiquitin protein ligase 1 (Hectd1) mRNA - that is likely a technical artifact since it is a cytosolic protein and the bait is extracellular - showed a dose-dependence for polyribosomes in

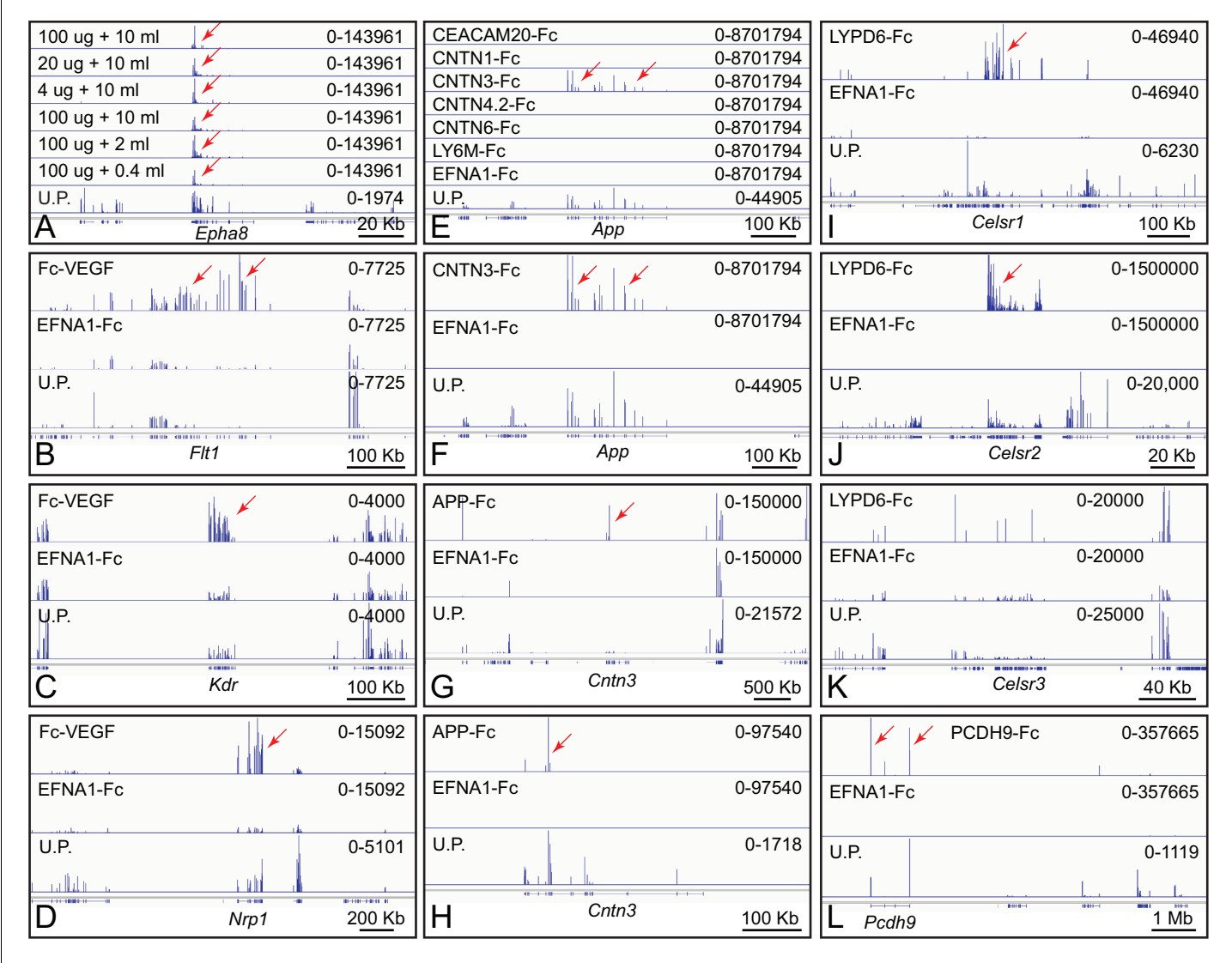

**Figure 2.** Genome browser images from ACAPseq of neonatal mouse brain polyribosomes captured with Fc fusion proteins. Genome browser images are presented as in *Figure 1B*. (**A**) Dilution of polyribosomes (in ug of RNA) and EFNA1-Fc bait (in ml of serum-free conditioned medium) showing the efficiency of Epha8 mRNA capture. (**B–D**) Fc-VEGF captured Flt1, Kdr, and Nrp1 mRNAs (red arrows). (**E–H**) CNTN3-Fc captured App mRNA and APP-Fc captured Cntn3 mRNA (red arrows). A subset of the panels in (**e**) are enlarged vertically in (**F**). (**G**) and (**H**) show the Cntn3 gene with different horizontal and vertical scales. The CNTN4 bait used here, CNTN4.2, is an isoform that lacks the three carboxy-terminal Fn domains. (**I–K**) LYPD6-Fc captured Celsr1 and Celsr2 mRNA (red arrows), but not Celsr3 mRNA. The several localized peaks of RNAseq reads in the Celsr3 region do not match the pattern of reads in the unselected polyribosome sample, and likely reflect artifacts of library amplification. (**L**) PCDH9-Fc captured Pcdh9 mRNA. The Pcdh9 gene spans ~1 Mb.

DOI: https://doi.org/10.7554/eLife.40982.003

The following figure supplement is available for figure 2:

**Figure supplement 1.** The capture of polyribosomes displaying nascent EPHA receptors by EFNA1-Fc bait is robust to variations in ACAPseq conditions.
DOI: https://doi.org/10.7554/eLife.40982.004

the same experiment, implying a lower affinity interaction with the EFNA1 bait (*Figure 2—figure supplement 1F*). Based on these results, standard ACAPseq reactions utilized ~5 – 20 mls of serum-free conditioned medium (corresponding to ~10 ug of bait-Fc), and 40 ug of polyribosomes. We also routinely included an EFNA1-Fc bait in each group of ACAPseq reactions as an internal control, and this control is included in each genome browser image.

ACAPseq reactions were performed with neonatal mouse brain polyribosomes with 92 baits derived from 87 genes coding for diverse cell surface and secreted proteins, including known or putative ligands, receptors, and cell adhesion proteins (*Supplementary file 1*; some baits are derived from different spliced isoforms of the same gene). All of the extracellular bait domains were fused to the amino-terminus of Fc. For VEGF, we also prepared a fusion to the carboxy-terminus of Fc (*Lo et al., 1998*), as our previous work had established that VEGF fusions to the carboxy-terminus of alkaline phosphatase were effective in receptor binding (*Wang et al., 2012*). Below, we describe the primary data for six bait-target combinations (summarized in *Supplementary file 2*). The interactions described in paragraphs (1), (2), and (5) have been reported previously; the interactions described in paragraphs (3), (4), and (6) are novel.

1. Fc-VEGF (the VEGF-A isoform) captured Flt1 (Vegfr1), Kdr (Vegfr2), and Nrp1 (Neuropilin1) mRNAs, which code for its known receptors and co-receptor, respectively (*Figure 2B–D*; *Koch and Claesson-Welsh, 2012*). In the whole brain polyribosome preparation, Flt1, Kdr, and Nrp1 mRNAs were present at abundances of 0.0009%, 0.0014%, and 0.01%, respectively, and were enriched 30-fold, 8-fold, and 8-fold by Fc-VEGF capture. Nrp2 mRNA, encoding a second co-receptor, was present at an abundance of 0.014%, but was only enriched 2.4-fold. The low abundance of Flt1 and Kdr mRNAs likely reflects their expression predominantly in vascular endothelial cells, a minor cell population in the brain. Fc-VEGF did not capture Flt4 mRNA, which codes for VEGFR3, a receptor that does not bind to VEGF-A, and which was present at an abundance of 0.0002% in whole brain polyribosomes.

2. Contactins are GPI anchored proteins consisting of six immunoglobulin (Ig) domains followed by four fibronectin (Fn) domains (*Zuko et al., 2011*). Using four of the six Contactin family members as baits (CNTN1, CNTN3, CNTN4.2, and CNTN6), Alzheimer's precursor protein (App) mRNA was captured specifically by CNTN3-Fc (*Figure 2E and F*). App mRNA is present in brain polyribosomes at an abundance of 0.080% and it was enriched 420-fold by ACAPseq. By contrast, APP-Fc captured Cntn3 mRNA (starting abundance 0.002%) with an enrichment of only 11-fold (*Figure 2G and H*). These interactions were previously reported by *Osterfield et al. (2008)*.

3. LY6/PLAUR domain containing 6 (LYPD6) is a single-domain GPI-anchored protein that interacts with nicotinic acetylcholine receptors and the WNT co-receptor LRP6 (*Özhan et al., 2013*; *Arvaniti et al., 2016*). LYPD6-Fc captured Cadherin EGF LAG seven-pass G-type receptor 1 (Celsr1) mRNA and Celsr2 mRNA, but not Celsr3 mRNA (*Figure 2I–K*). The abundances of Celsr mRNAs in brain polyribosome are: 0.0011% (Celsr1), 0.012% (Celsr2), and 0.0045% (Celsr3). ACAPseq enrichments were 250-fold (Celsr1), 475-fold (Celsr2), and 0.9-fold (Celsr3). CELSRs are ~3000 amino acids in length with seven transmembrane domains and a large amino-terminal extra-cellular domain. CELSR1 forms homophilic complexes between neighboring epithelial cells, and its loss disrupts planar cell polarity signaling. CELSR2 and CELSR3 regulate ciliogenesis, neuronal migration, and axon guidance (*Tissir and Goffinet, 2013*).

4. Protocadherin-9 (PCDH9) is one of five members of the delta-1 subgroup of nonclustered protocadherins. Its ECD consists of seven tandem cadherin domains. Like many nonclustered protocadherins, it is expressed in a region-specific pattern in the developing cerebral cortex (*Kim et al., 2007*). As seen in *Figure 2L*, PCDH9-Fc captures its own polyribosomes, implying a homophilic mode of binding. Pcdh9 mRNA is present in brain polyribosomes at an abundance of 0.019% and it was enriched 300-fold by ACAPseq.

5. Latrophilins (LPHNs; encoded by Adgrl genes) have seven transmembrane-domains and a large amino-terminal extra-cellular domain (*Silva and Ushkaryov, 2010*). As noted above in the context of *Figure 1B*, LPHNs bind to FLRT family members. *Figure 3A–D* shows genome browser images from ACAPseq with LPHN1 and LPHN3 baits, which show nearly identical capture specificities for Flrt family members, with Flrt1 mRNA enriched 2.5-fold, and Flrt2 and Flrt3 mRNA enriched 50-fold and 100-fold, respectively. The abundances of Flrt1, Flrt2 and Flrt3 mRNAs in brain polyribosomes are 0.0021%, 0.012%, and 0.0037%, respectively. These interactions have been reported by multiple investigators (e.g. *Jackson et al., 2016*).

6. Leucine-rich repeats and transmembrane domain 2 (LRTM2) is a 370 amino acid protein of unknown function. LRTM2-Fc captured mRNAs coding for the three members of the Tenascin family of extracellular matrix proteins, TNC, TNN, and TNR (*Figure 3E–H*). Tnc, Tnn, and Tnr mRNAs are present in brain polyribosomes at abundances of 0.0045%, 0.00002%, and 0.0047%, respectively and were enriched 215-fold, 40-fold, and 25-fold. Tenascins have been implicated in cell adhesion and motility in a wide variety of tissues, including the brain (*Chiquet-Ehrismann and Tucker, 2011*).

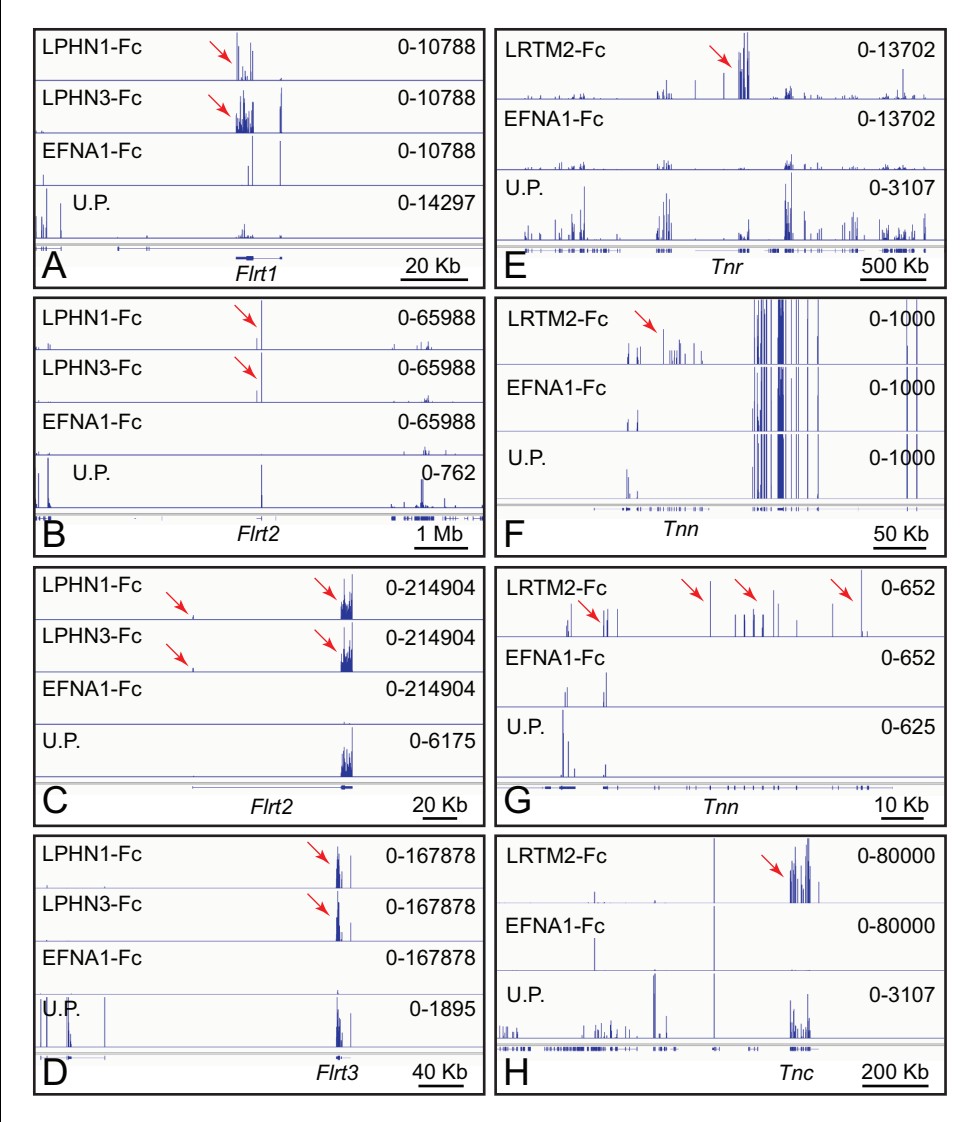

**Figure 3.** ACAPseq with LPHN1-Fc, LPHN3-Fc, and LRTM2-Fc baits. (A–D) Flrt1, Flrt2, and Flrt3 mRNAs captured with LPHN1 and LPHN3 baits (red arrows). (B) and (C) are centered on the Flrt2 gene but have different horizontal and vertical scales. (E–H) Tnr, Tnn, and Tnc mRNAs captured with the LRTM2 bait (red arrows). (F) and (G) show the Tnn gene with different horizontal and vertical scales. In (H), the pile up of many reads at a single location in the LRTM2, EFNA1, and total polyribosome tracks is presumed to be a sequencing and/or alignment artifact. Genome browser images from ACAPseq of neonatal mouse brain polyribosomes, as shown in *Figure 1B*.
DOI: https://doi.org/10.7554/eLife.40982.005

The following figure supplement is available for figure 3:

**Figure supplement 1.** Presumptive artifactual and non-specific bait-target interactions in ACAPseq.
DOI: https://doi.org/10.7554/eLife.40982.006

For several dozen mRNAs, we observed non-specific capture by all or nearly all baits, an effect that we ascribe to a general 'stickiness' of the corresponding nascent polypeptides (*Figure 3—figure supplement 1A–C*). Somewhat less promiscuous binding was observed with some other mRNAs, for example Plec and Golgb1 mRNA (*Figure 3—figure supplement 1D and E*). Golgb1 codes for Giantin, a ~ 3300 amino acid member of the Golgin family of coiled-coil-containing transmembrane proteins that localizes to and maintains the structure of the Golgi stack (*Koreishi et al., 2013*). Given

Giantin's localization along the secretory pathway, its association with diverse ECDs may be biologically meaningful.

Finally, some ACAPseq interactions that appear to be highly specific are likely to be biologically irrelevant. An example is seen in the capture of Hectd1 mRNA by the EFNA1 bait (*Figure 3—figure supplement 1F*). As HECTD1 is a cytoplasmic E3 ubiquitin ligase and the EFNA1 bait is derived from the ECD, these two proteins are on opposite sides of a lipid bilayer at all stages of their existence. We presume that there is no selective pressure to eliminate the chance complementarity of the two proteins' surfaces since they never interact under normal circumstances. As noted above, the interaction between the EFNA1 bait and nascent HECTD1 on polyribosomes appears to be weaker than the interaction between the EFNA1 bait and nascent EPHA receptors, as judged by the reduced efficiency of Hectd1 capture with lower polyribosome input.

## Genome-wide visualization of ACAPseq specificity

To visualize ACAPseq data on a genome-wide scale, we have plotted normalized RNAseq read counts (in the form of transcripts per million; TPM) for each of the ~25,000 protein coding genes in 2-dimensional scatter plots, with reads from the internal control EFNA1-Fc bait on the x-axis and reads from each of the other baits on the y-axis. We refer to these plots as ACAPseq scatter plots (ASPs).

As seen in *Figure 4*, most data points in a typical ASP cluster at or near the origin, representing (1) mRNAs that are present in the starting polyribosome sample at very low abundance, and (2) mRNAs of all abundances that are not captured (or not captured efficiently) by either bait. Several dozen data points representing mRNAs coding for non-specifically 'sticky' nascent polypeptides lie along the 45 degree line, indicative of binding to Protein-G beads and/or similar levels of binding to many bait-Fc fusion proteins. Data points with a high x:y ratio represent mRNAs specifically captured by EFNA1-Fc. Data points with a high y:x ratio represent mRNAs specifically captured by the bait of interest. The former group codes for multiple EPHA family members, as well as HECTD1 and OS9, a lectin that localizes to the endoplasmic reticulum and binds partially folded proteins (*Seidler et al., 2014*).

For the nine ASPs shown in *Figure 4*, most of the data points representing 'real' bait-target interactions were clearly separate from the bulk of the data points. For most of these targets there was a non-zero level of capture with control EFNA1 bait and a 10-fold to 500-fold greater capture efficiency with the relevant bait. In addition to the 'real' targets, a variety of highly specific but likely artifactual interactions were revealed by the ASPs, including intracellular proteins encoded by Poly-bromo-1 (Pbrm1) for FLRT3-Fc (*Figure 4A*), mitochondrial Leucyl-tRNA synthetase-2 (Lars2) for LPHN1-Fc (*Figure 4B*), Small nuclear ribonucleoprotein N (Snrpn) for Fc-VEGF (*Figure 4C*), and Cyclin-dependent kinase 11b (Cdk11b) for APP-Fc (*Figure 4F*). These targets can be eliminated by applying a filter to remove cytoplasmic and nuclear proteins. ASPs provide a rapid visual assessment of the specificity of capture for multiple target mRNAs, as seen for example in *Figure 4G*, for the capture of Celsr1 and Celsr2 but not Celsr3 by LYPD6-Fc.

Close inspection of the ASPs in *Figure 4* shows the importance of minimizing within-batch variability. *Figure 4C and I* show tight clustering of non-specific interactions along the 45-degree line, facilitating the separation of data points representing Flt1, Kdr, and Nrp1 mRNAs from the bulk of the data points (*Figure 4C*), although this clustering was insufficiently tight to permit a separation of Tnn mRNA from the bulk of the data points in the LRTM2 ASP (*Figure 4I*). In the other ASPs in *Figure 4*, the scatter in the data is larger than in *Figure 4C and I*. Thus, in the ASP for APP-Fc, the variability in non-specific interactions overlaps the data point representing Cntn4 mRNA (*Figure 4F*).

## ACAPseq reproducibility

We have assessed the reproducibility of ACAPseq by comparing twelve independent EFNA1 ACAPseq reactions that were embedded as positive controls in each batch of ACAPseq reactions (*Figure 5*). This analysis is particularly informative because it includes multiple validated EFNA1 targets (i.e., multiple EPHA receptors) encompassing a wide range of mRNA abundances and capture efficiencies. *Figure 5A and B* compare two pairs of EFNA1 ACAPseq results: one pair from within a batch and a second (and representative) pair from two different batches. The Pearson correlations for the within-batch pair are 0.899 for all data points and 0.998 for the Epha and Ephb mRNAs, and

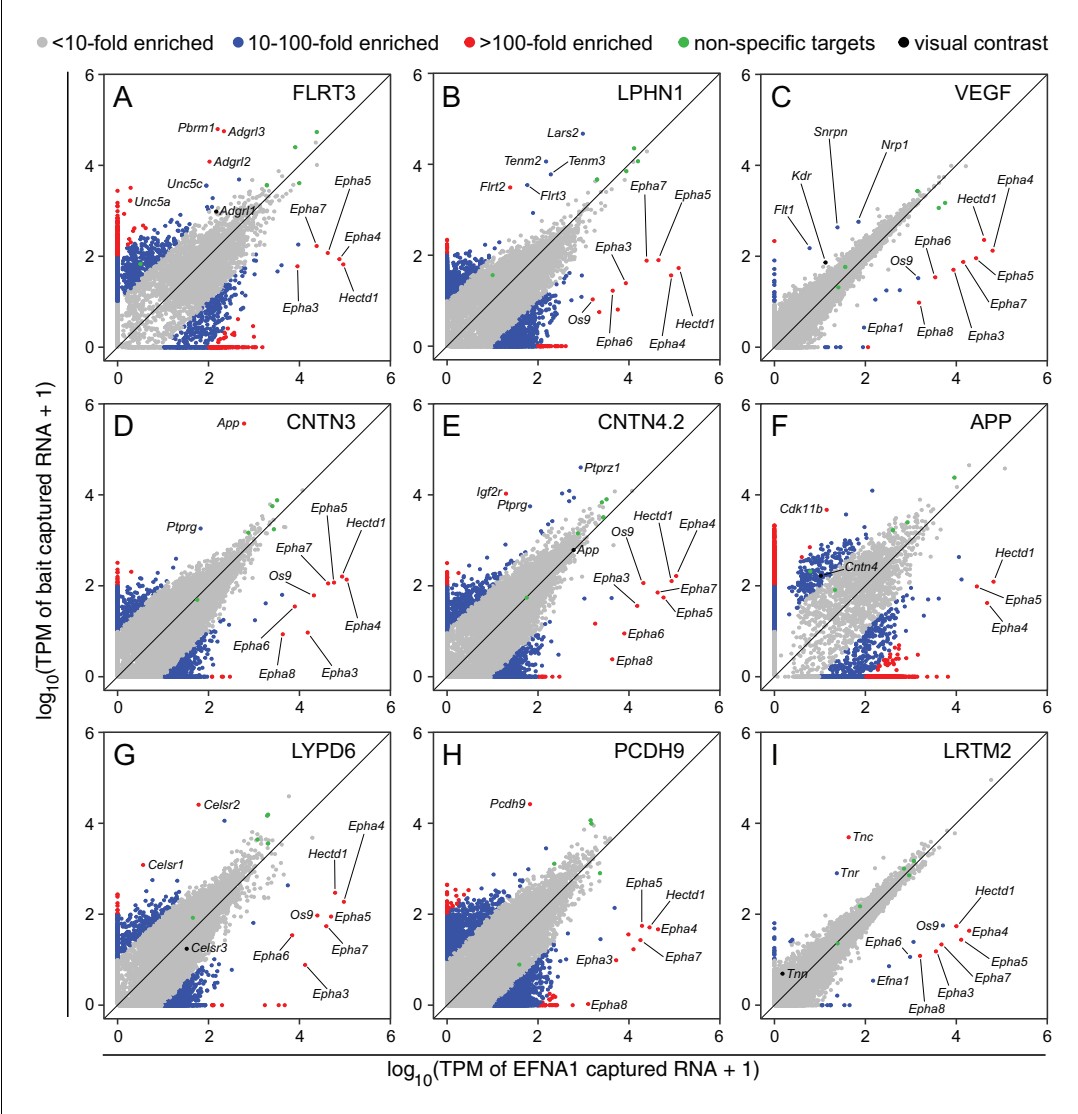

**Figure 4.** Two-dimensional scatterplots for genome-wide visualization and quantification of ACAPseq. Log$_{10}$ TPM counts are plotted for each of nine baits (indicated in each panel) vs. EFNA1 from the same experiment. Multiple Epha receptor mRNAs, known EFNA1 targets, are captured by the EFNA1 bait. Data points representing mRNAs with different fold enrichments are indicated by grey, blue, and red dots. Five polyribosome targets – Dbn1, Golgb1, Myo10, Plec, Trip11 – with promiscuous binding (presumably derived from 'sticky' nascent polypeptides; *Figure 3—figure supplement 1*) are represented by green dots.

DOI: https://doi.org/10.7554/eLife.40982.007

the correlations for the between-batch pair are 0.947 for all data points and 0.988 for the Epha and Ephb mRNAs. Both comparisons preserve the rank order of the seven most efficiently captured Epha mRNAs, which span three orders of magnitude in TPM values. Across the twelve EFNA1 ACAPseq reactions, a comparison of Epha capture efficiencies – that is TPM counts for mRNAs from nine captured Epha genes vs. two independent neonatal brain polyribosome samples - shows that (1) captured read counts correlate with starting read counts, (2) the rank order of capture efficiency for Epha family members is consistent across experiments, and (3) different experiments differ in the overall capture efficiencies across all Epha targets (*Figure 5C*).

## Quantifying ACAPseq performance

To quantify ACAPseq performance, we calculated the sensitivity, specificity, and false discovery rate (FDR) of target detection as a function of two threshold parameters: (1) absolute TPMs for captured

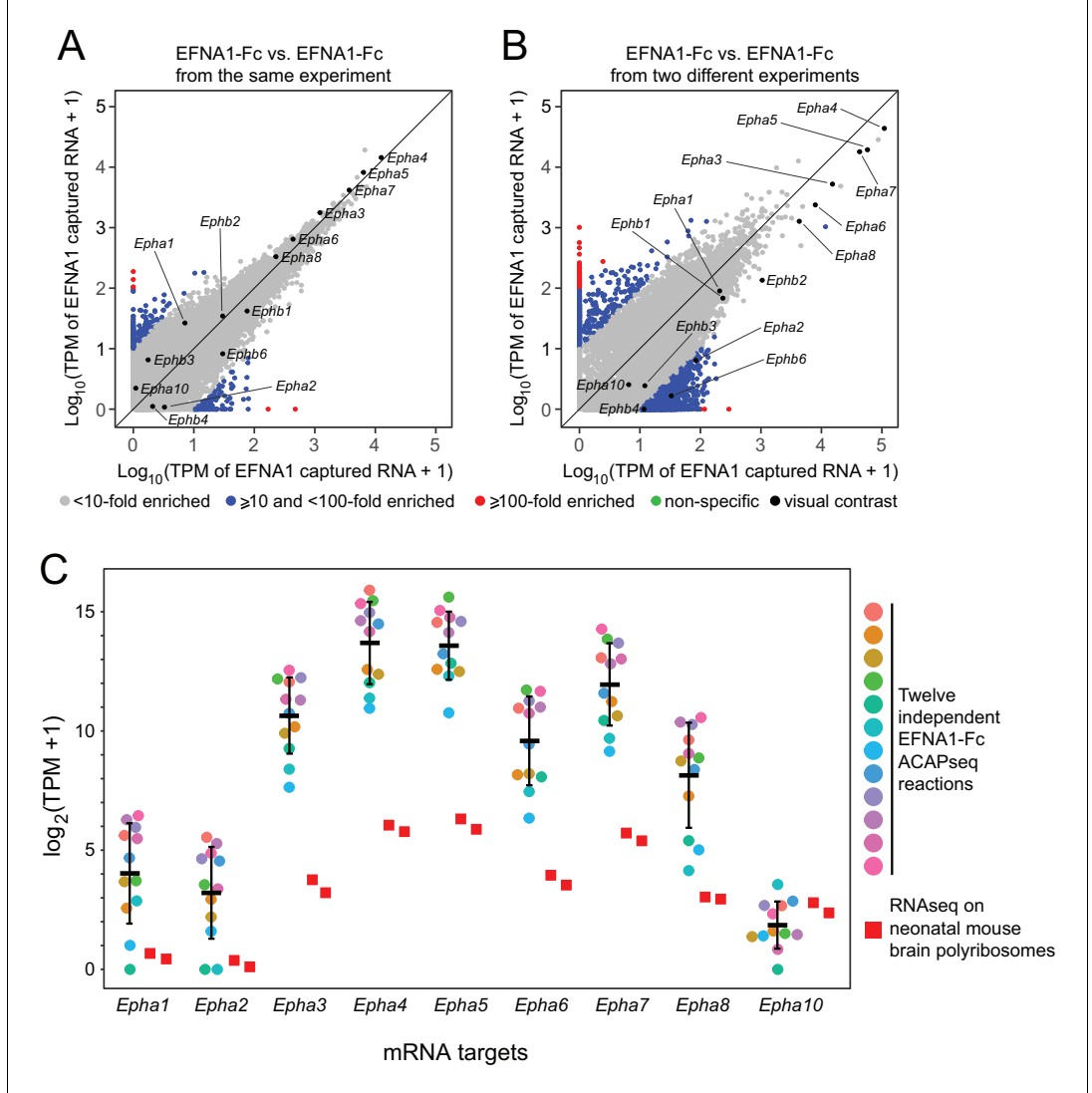

**Figure 5.** Reproducibility of ACAPseq assessed with EFNA1 capture of multiple EPHA receptor mRNAs. (**A,B**) ASPs of two EFNA1 ACAPseq reactions from (**A**) the same experiment and (**B**) two independent experiments. (**C**) TPMs for two independent starting neonatal mouse brain polyribosome samples (r = 0.984 for all protein-coding genes) and 12 independent EFNA1 ACAPseq reactions for each of nine Epha mRNAs. The vertical axis spans a 32,000-fold range. Error bars, mean ±S.D. of the $\log_2$ (TPM +1) values.

DOI: https://doi.org/10.7554/eLife.40982.008

The following figure supplements are available for figure 5:

**Figure supplement 1.** Sensitivity, specificity, and FDR for ACAPseq with Fc-VEGF and EFNA1-Fc baits.

DOI: https://doi.org/10.7554/eLife.40982.009

**Figure supplement 2.** Quantifying ACAPseq bait-target pairs with a defined threshold.

DOI: https://doi.org/10.7554/eLife.40982.010

mRNAs, and (2) fold-enrichment of captured mRNAs relative to either mRNA abundance in the starting whole brain polyribosome sample or mRNA abundance following capture with multiple unrelated baits from the same experiment. For this analysis, we used representative ACAPseq experiments with EFNA1-Fc or Fc-VEGF baits, which included, respectively, seven and five unrelated baits. These two baits were chosen because each has multiple validated targets. For the fold enrichment relative to unrelated baits, a target was categorized as positive only if the threshold TPM ratio was exceeded for every comparison with the unrelated baits. To further enhance the signal-to-noise ratio, we included only mRNAs with TPM >0 in mouse brain polyribosomes and coding for membrane or

secreted proteins (N = 5537; *Uhlén et al., 2015*); https://www.proteinatlas.org/humanproteome/secretome).

For VEGF-A, there are four known binding partners: VEGFR-1/FLT1, VEGFR-2/KDR, NRP1, and NRP2. For EFNA1, the binding partners could maximally correspond to the nine EPH-A and five EPH-B family members encoded in the mouse genome. In fact, only nine of the 14 EPH family members have been shown to bind to EFNA1 (EPHA-A1, -A2, -A3, -A4, -A5, -A6 -A7, -A8, and -B2; *Noberini et al., 2012*). In *Figure 5—figure supplement 1*, calculations referring to nine or 14 true targets are shown outside or inside parentheses, respectively.

*Figure 5—figure supplement 1A* summarizes the sensitivity, specificity, and FDR calculations for TPM cut-offs of zero or ten, and fold-enrichment cut-offs of three or ten. For Fc-VEGF, a fold enrichment cut-off of ten appears to be too stringent, as this gave a sensitivity of zero or 0.25, depending on the other parameters. With total brain polyribosomes as the control, a fold-enrichment cut-off of three gave a sensitivity of 1.0 with an FDR ranging from 0.9 to 0.95. With irrelevant baits as the control, the sensitivity was 0.75 with an FDR ranging from 0 to 0.69. Across the parameter values explored here, the specificity varied from 0.987 to 1.0. For EFNA1-Fc, ACAPseq performance was relatively insensitive to this range of parameter values, with sensitivities of 0.89 – 1.0 (for nine true EPH targets), specificities of 0.977 – 0.997, and FDRs of 0.64 – 0.93. The TPM cut-off could be set as high as 60 with little effect on EFNA1-Fc sensitivity and specificity.

*Figure 5—figure supplement 1B* shows the enrichment ratio of specific bait vs. irrelevant baits for Fc-VEGF using a TPM cut-off of ten and a fold-enrichment of three (left) and for EFNA1-Fc using a TPM cutoff of zero and a fold enrichment of ten (right). For EFNA1-Fc, we can maximize the signal-to-noise ratio by setting the fold-enrichment threshold at $\log_{10}$ ~1.3 (red line in *Figure 5—figure supplement 1B*, right), which would reduce the FDR to 0.11. *Figure 5—figure supplement 1C* compares the enrichment distribution for all mRNAs in excess of the indicated TPM cut-off vs. the enrichments for true targets of VEGF (left pair of panels) and EFNA1 (right pair of panels). It is apparent that the ACAPseq enrichment ratios for most of the true targets are many standard deviations from the mean of the distribution.

Taken together, these examples show the high performance of ACAPseq, but they also imply that no single set of cut-off parameters are optimal for all baits, presumably due to differences in the starting abundance of target mRNAs, the affinities of bait-target binding, and the abundance of mRNAs coding for targets that bind nonspecifically to the bait.

## Thresholds for scoring ACAPseq, and a comparison with existing PPI databases

Based on the preceding analysis of ACAPseq performance, a threshold for target interaction was set with the following four criteria: (1)>35 captured TPMs, (2) captured TPMs > 3 x the maximal number of TPMs captured with any of the other baits used in the same experimental cohort, (3) captured TPMs > 3 x the TPMs in the unselected polyribosomes, and (4) the target is a secreted or membrane protein (as designated by https://www.proteinatlas.org/humanproteome/secretome). With these criteria, we found a total of 1207 supra-threshold targets for the 92 baits (*Supplementary file 4*). The distribution of targets per bait is highly skewed (*Figure 5—figure supplement 2A*), with six baits showing >50 targets per bait and three baits showing >100 targets per bait, suggesting that the small fraction of baits that had a large number (e.g. >30) targets may be nonspecifically sticky. Fifty-two baits (57%) showed 1 – 10 targets per bait and 15 baits (16%) showed no targets. The values for the bait vs. control RNAseq read-counts cover a wide range (*Figure 5—figure supplement 2B*), suggesting the potential utility of a more sophisticated algorithm that assigns a weight to each bait-target pair as a function of the bait vs. control RNAseq read-count ratio.

Among the 86 baits derived from different genes, 24 (28%) capture one or more targets that (1) are biologically plausible (i.e. the target is an extracellular or transmembrane protein with large extracellular domains), (2) have an ACAPseq signal that is well in excess of non-specific binding, and (3) are experimentally validated in the present study, were previously reported, and/or are related to a previously reported target (e.g. is a member of the same protein family) (*Supplementary file 1*). Altogether, there were 64 bait-target pairs in this category. Additional validation experiments will be required to determine the fraction of the other members of the 1207 supra-threshold targets that represent legitimate bait-target pairs.

A comparison of ACAPseq with two large public data bases of PPIs – BioGRID (*Stark et al., 2006*; http://www.thebiogrid.org) and IntAct (*Orchard et al., 2014*; https://www.ebi.ac.uk/intact/) - show a surprisingly low degree of overlap among candidate PPIs. Among baits that were listed in these databases and tested in the present study (47 for BioGRID and 43 for IntAct), there were only eight bait-target pairs in common between each database and the ACAPseq dataset (*Figure 5—figure supplement 2C*). This low degree of overlap (both in absolute numbers and relative to the total number of supra-threshold bait-target pairs) likely reflects (1) the incomplete coverage of published data in the public databases, and (2) the present lack of validation for many bait-target pairs. An example of the first of these factors is seen in the coverage of LPHN/ADGRL, UNC5, and FLRT family members, all of which interact with each other, as seen in the structure of the 2:1:1 complex of LPHN3/ADGRL3, UNC5D, and FLRT2 (*Jackson et al., 2016*). Among targets for FLRT and LPHN family members, no UNC5 family interactions are listed in BioGRID and only a single interaction (with UNC5B) is listed in IntAct. ACAPseq detects interactions between FLRT1 and UNC5A and C, and between FLRT3 and UNC5A, B, C, and D (*Supplemental file 1*).

## ACAPseq specificity across members of a large protein family

As ACAPseq interrogates all of the polyribosomes in a population, a single experiment can potentially define the specificity of bait binding across many members of a protein family. In particular, for protein families where (1) amino-terminal sequences in the target proteins mediates bait binding and (2) polyribosomes displaying nascent polypeptides from one family member can bind to the bait, then it is likely that nascent polypeptides from other family members will be similarly available for binding. We have applied this logic to assess the potential for interactions between PCDH9 and the members of the classical cadherin, non-clustered protocadherin, and clustered protocadherin families that are expressed in the neonatal mouse brain.

The ASPs in *Figure 6A* compare PCDH9 vs. EFNA1 ACAPSeq for cadherin and non-clustered protocadherin family members (left panel) and for clustered protocadherin family members (right panel). *Figure 6B* compares TPMs from PCDH9-Fc captured mRNAs vs. neonatal mouse brain polyribosome mRNAs for individual cadherin and protocadherin subfamilies. Pcdh9 mRNA shows a ~ 100 fold higher ratio of PCDH9-Fc vs. EFNA1-Fc capture compared to other cadherin and protocadherin family members. Captured Pcdh9 mRNA also shows a ~ 100 fold enrichment over its starting mRNA abundance. None of the other cadherin or protocadherin mRNAs show more than several-fold enrichment over their starting abundances, and the vast majority show either no change or a relative depletion following PCDH9-Fc capture. *Figure 6B* also shows that Pcdh9 mRNA is one of the more abundant cadherin/protocadherin mRNAs in neonatal mouse brain, but mRNA abundance should not affect either the capture efficiency per polyribosome or the ratio of PCDH9-Fc to EFNA1-Fc capture efficiencies, which are, presumably, intrinsic properties of the individual polyribosomes and their interactions with the bait-Fc beads. If we assume that potential binding interactions between PCDH9 and other cadherin/protocadherin targets involve cadherin domains that are close to the amino-terminus and are therefore accessible in the context of the nascent polypeptide, then these data imply that PCDH9 interactions with cadherin/protocadherin family members in the neonatal mouse brain are largely limited to homophilic binding.

As noted above, we compared VEGF-Fc and Fc-VEGF baits in ACAPseq. In each of two independent experiments, Fc-VEGF gave consistently higher capture efficiencies for Flt1, Kdr, and Nrp1 (*Figure 6—figure supplement 1*). For Nrp1 polyribosomes, VEGF-Fc produced no enrichment over the starting polyribosome sample. These data emphasize the utility of exploring different configurations of bait-partner protein fusions when designing ACAPseq probes.

## Comparison of ACAPseq with brain vs. testis polyribosomes

One measure of the utility of a PPI discovery platform is its performance in identifying bone fide targets in high complexity starting samples. In vertebrates, the highest complexity transcriptomes are found in brain and testis. RNAseq data indicate that ~ 16,000 (~70%) of protein-coding genes are expressed in testis, most likely as a consequence of a genome-wide process that resets the chromatin landscape (*Soumillon et al., 2013*). If most or all of these testis transcripts are translated, then testis-derived polyribosomes might serve as an especially useful starting point for ACAPseq, since the majority of the proteome would be displayed as nascent polypeptides on this polyribosome

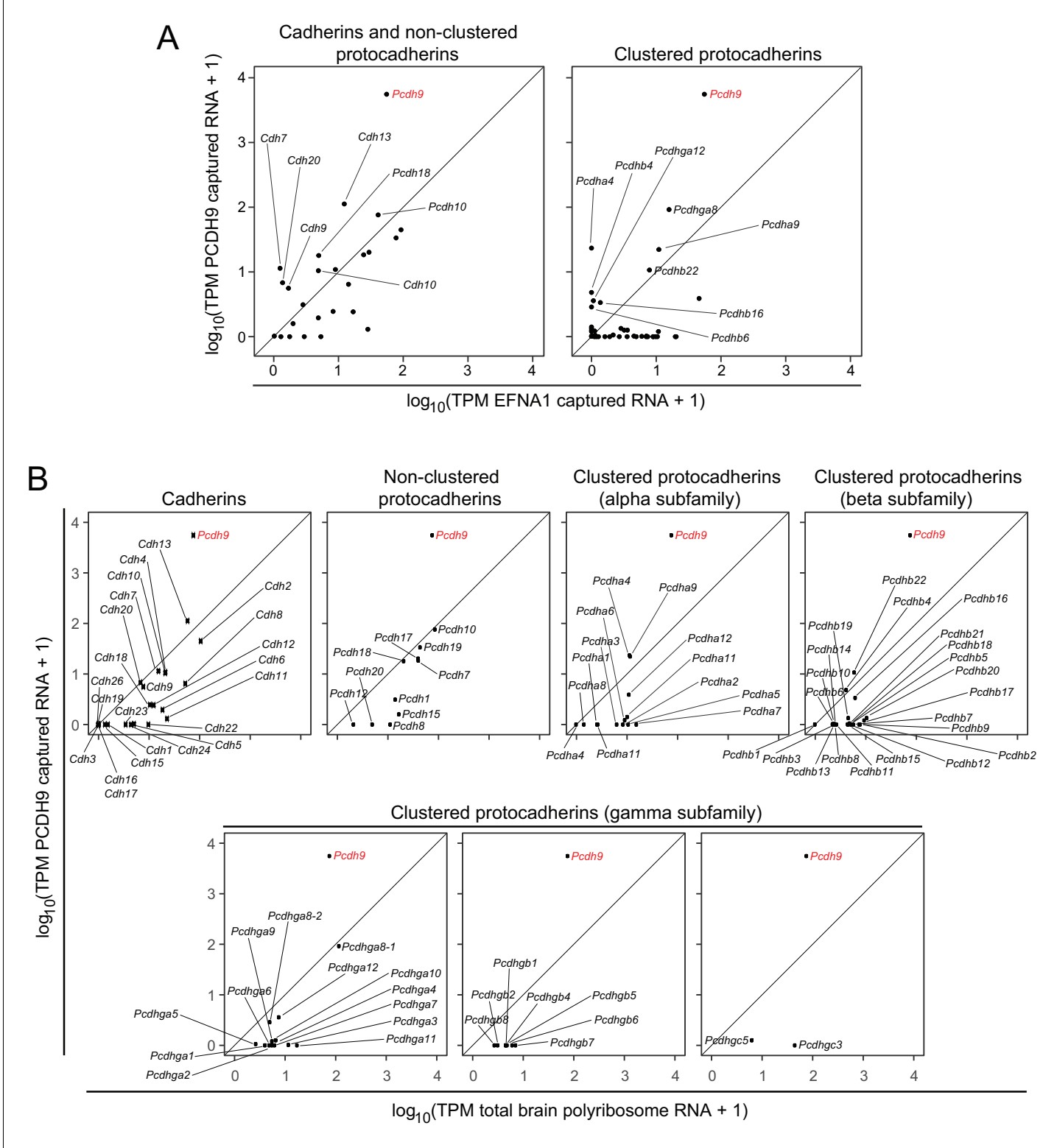

**Figure 6.** ACAPseq with PCDH9-Fc is highly selective for Pcdh9 polyribosomes relative to polyribosomes from other cadherin and protocadherin family members. (**A**) ASPs showing PCDH9 vs. EFNA1 capture of mRNA coding for cadherin and non-clustered protocadherin family members (left) and clustered protocadherin family members (right). Only those mRNAs that were captured more efficiently by PCDH9-Fc compared to EFNA1-Fc are labeled. (**B**) ASPs showing TPMs for PCDH9 captured mRNA vs. total neonatal mouse brain polyribosome mRNA coding for cadherins, non-clustered

*Figure 6 continued on next page*

*Figure 6 continued*

protocadherins, and subfamilies of clustered protocadherins. Pcdh9 mRNA is captured far more efficiently than any other cadherin/protocadherin family member (red label).

DOI: https://doi.org/10.7554/eLife.40982.011

The following figure supplement is available for figure 6:

**Figure supplement 1.** Fc-VEGF is a more effective bait than VEGF-Fc.

DOI: https://doi.org/10.7554/eLife.40982.012

population. We note that if some genes exhibit sex-limited expression in testis, then this starting material could be systematically biased against genes that are expressed specifically in females.

When EFNA1-Fc was incubated with adult mouse testis polyribosomes, multiple Epha targets were captured with a high signal-to-noise ratio (*Figure 7A*), even though most of the Epha targets are present at >10 fold lower abundance in testis compared to brain (*Figure 7B*). In a standard ASP comparison with brain ACAPseq, the TPMs of captured testis transcripts were observed to be >10 fold lower than their brain counterparts, consistent with their lower starting abundances (*Figure 7C*). By plotting the ratio of captured vs. starting RNA abundances, 7/8 Epha targets enriched in brain ACAPseq were observed to be enriched above the bulk of the transcripts in testis ACAPseq, with good agreement between the normalized brain and testis capture efficiencies (*Figure 7D*). The only exception to this pattern was Epha6, which is not detectably enriched in the testis sample. These data suggest that testis polyribosomes could serve as a useful starting material for ACAPseq.

## Representation of alternative splicing events in ACAPseq data

One potentially advantageous feature of ACAPseq is that it surveys protein products representing the full range of mRNA transcript diversity in the starting polyribosome population. Transcript diversity can be generated by RNA splicing, RNA editing, or other post-transcriptional processes, or it can be encoded in the genome, as seen with single nucleotide variants (SNVs). *Figure 8A and B* shows an example of alternative splicing in the Cntn4 gene, in which inclusion of an alternate exon generates a short protein isoform that is missing the three carboxy-terminal fibronectin domains (GenBank NM_173004; available at https://www.ncbi.nlm.nih.gov/nuccore/NM_173004); skipping the alternate exon generates a long isoform that encodes the full-length protein (GenBank NM_001109749; available at https://www.ncbi.nlm.nih.gov/nuccore/NM_001109749.1). Both isoforms code for a protein that includes the previously defined PTPRG binding site in the second and third immunoglobulin domains (*Nikolaienko et al., 2016*). Using a PTPRG-Fc bait, transcripts corresponding to both long and short isoforms were captured by ACAPseq (*Figure 8E*). We note that the short isoform includes only eight amino acids that are not present in the long isoform, which could make its detection by mass spectrometry challenging. By contrast, from the polyribosomes captured by PTPRG-Fc, dozens of independent RNAseq reads are derived from the alternate exon.

An analogous result was obtained with ENA1-Fc capture of transcripts coding for long and short isoforms of EPHA7 (GenBank NM_010141 and NM_001122889, respectively). Both isoforms contain the entire ECD, but the short isoform is missing the cytoplasmic tyrosine kinase domain (*Figure 8C, D and F*).

## Validating LTRM2 and LYPD6 interactions

To independently assess the ACAPseq interaction between LRTM2 and TNR, one of three tenascin targets (*Figure 3E* and *Figure 4I*), epitope-tagged wild type (WT) TNR, a TNR derivative lacking the N-terminal trimerization domain and the three tandem EGF domains, and a TNR derivative lacking 7 of the nine tandem Fn domains were produced in transiently-transfected HEK/293 T cells, solubilized from cell lysates, captured with LRTM2-Fc, and visualized by immunoblotting (*Figure 8—figure supplement 1*). This experiment shows that the region encompassing the trimerization domain and the three tandem EGF domains is essential for LRTM2-Fc binding.

To assess the predicted interaction between LYPD6 and its CELSR targets, we examined the binding of a LYPD6-alkaline phosphatase (AP) fusion protein to live HEK/293 T cells expressing each of the three CELSR family members (*Figure 8—figure supplement 2*). This assay was used in lieu of binding assays with soluble proteins because our attempts to express the ~2000 amino acid CELSR

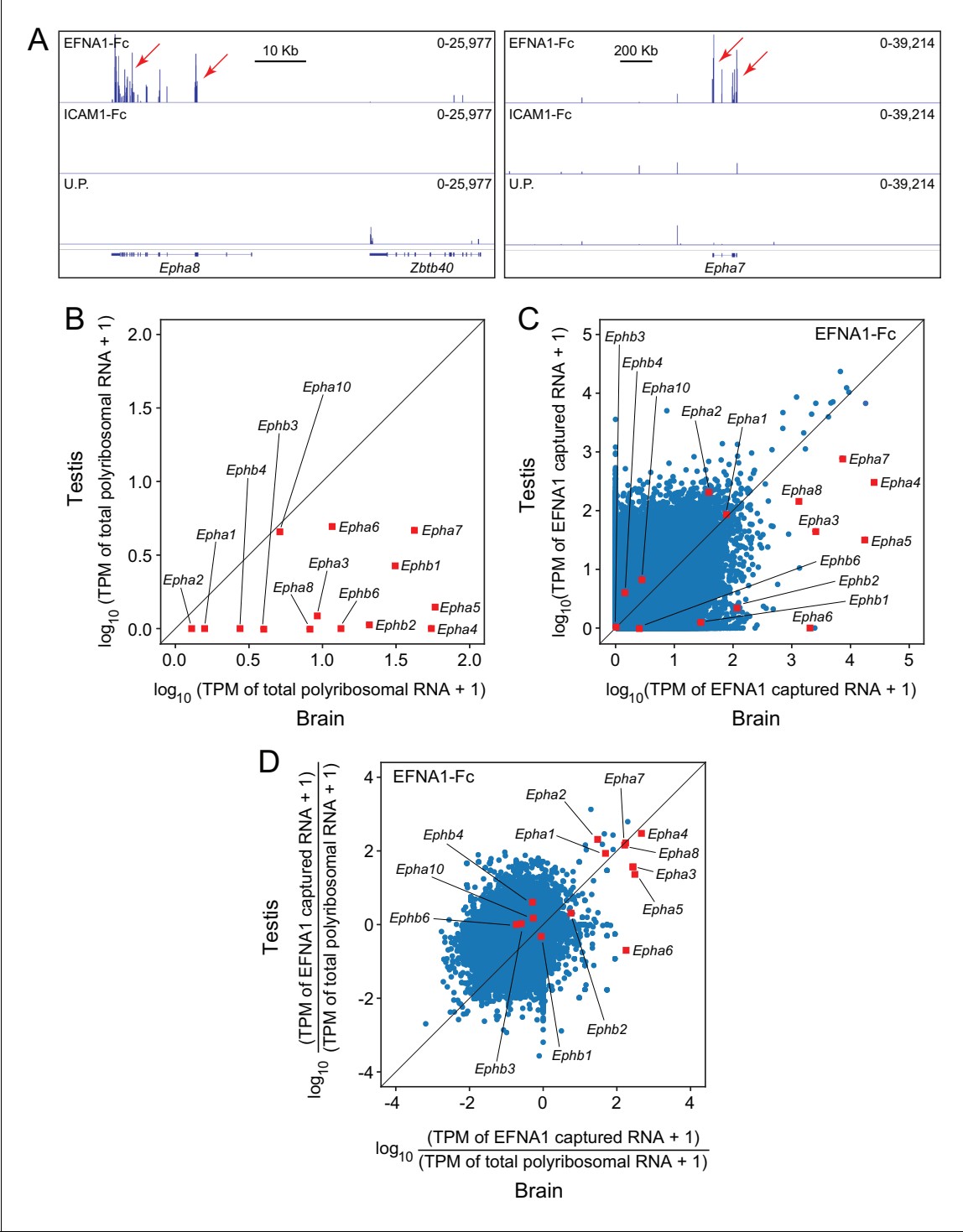

**Figure 7.** ACAPseq using polyribosomes from mouse testis. (A) Epha8 and Epha7 mRNAs captured with EFNA1-Fc baits (red arrows). Genome browser images, as shown in *Figure 1B*. (B) Abundances of transcripts coding for EPHA and EPHB family members in testis vs. brain total polyribosomes. (C) ASP of EFNA1-Fc captured from testis vs. brain polyribosomes. (D) Comparison of testis vs. brain EFNA1-Fc ACAPseq, with read counts from each captured transcript normalized to the starting abundance of that transcript.

DOI: https://doi.org/10.7554/eLife.40982.013

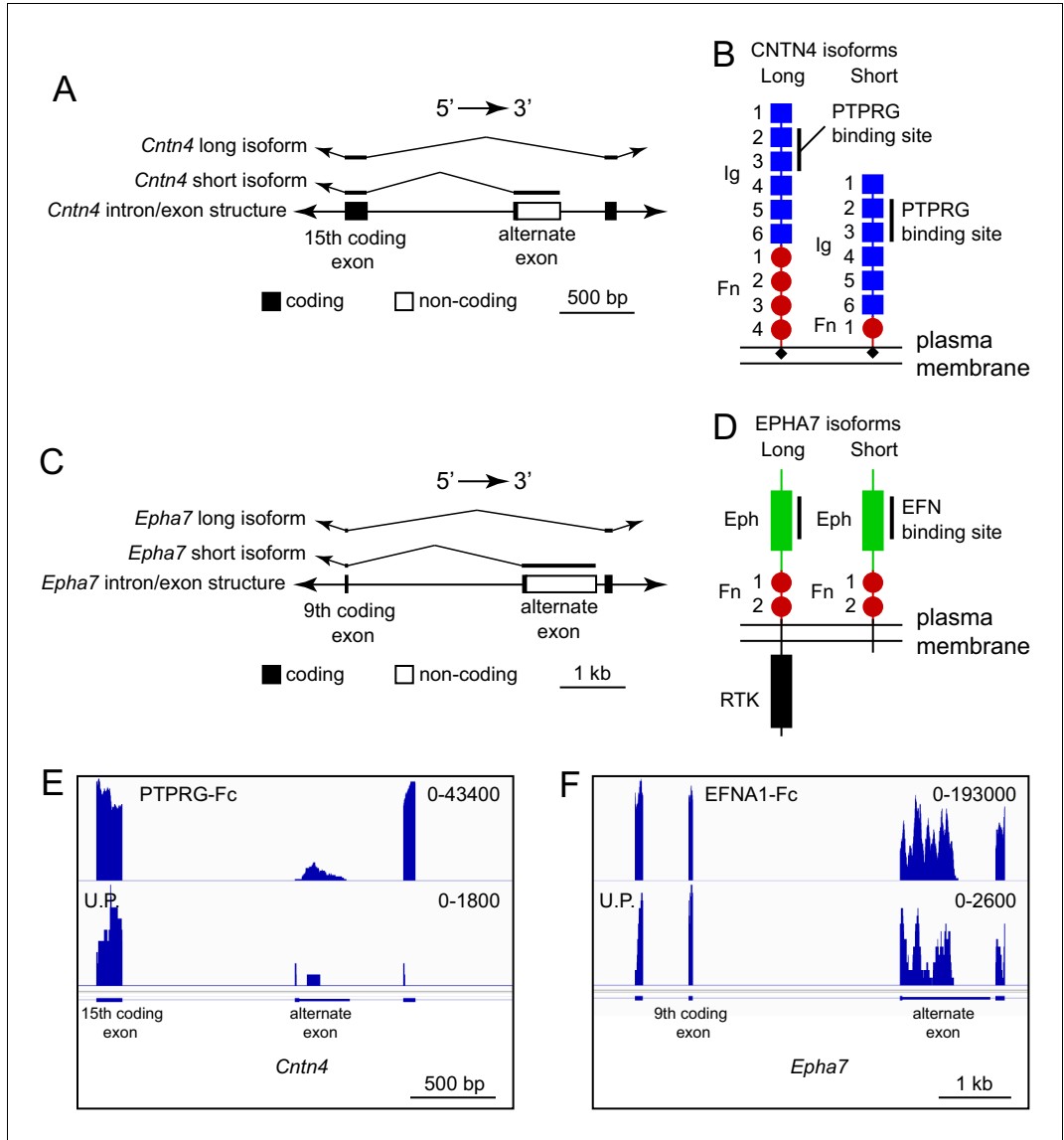

**Figure 8.** ACAPseq capture of alternately spliced isoform transcripts of Cntn4 and Epha7. (**A**) Cntn4 gene structure showing the location of an alternate exon 3' of coding exon 15. The alternate exon has a short open reading frame followed by a 3' untranslated region. The short CNTN4 isoform is encoded by transcripts that end with the alternate exon and the long CNTN4 isoform is encoded by transcripts that skip the alternate exon. (**B**) Diagrams of the long and short CNTN4 isoforms showing the location of the PTPRG binding site in the second and third Ig domains (**Nikolaienko et al., 2016**). The short isoform is missing Fn domains 2 – 4. Both isoforms are GPI-anchored plasma membrane proteins. (**C**) Epha7 gene structure showing the location of an alternate exon 3' of coding exon 9. The alternate exon has a short open reading frame followed by a 3' untranslated region. The short EPHA7 isoform is encoded by transcripts that end with the alternate exon and the long EPHA7 isoform is encoded by transcripts that skip the alternate exon. (**D**) Diagrams of the long and short EPHA7 isoforms showing the location of the EFNA1 binding site in the Eph domain (**Himanen et al., 2009**). The short isoform has a small intracellular tail in place of the receptor tyrosine kinase domain. (**E,F**) Genome browser images from ACAPseq of neonatal mouse brain polyribosomes, as shown in **Figure 1B**, with PTPRG-Fc (**E**) and EFNA1-Fc (**F**) as baits. PTPRG-Fc captured Cntn4, and EFNA1-Fc captured Epha7. The regions shown correspond to the diagrams in (**A**) and (**C**).

DOI: https://doi.org/10.7554/eLife.40982.014

The following figure supplements are available for figure 8:

**Figure supplement 1.** The amino-terminal domain(s) of TNR bind LRTM2-Fc.
DOI: https://doi.org/10.7554/eLife.40982.015

**Figure supplement 2.** LYPD6-AP binds to CELSR1 and CELSR2 but not to CELSR3 on the surface of living cells.
DOI: https://doi.org/10.7554/eLife.40982.016

amino-terminal ECDs as secreted fusion proteins were unsuccessful. As seen in *Figure 8—figure supplement 2B*, LYPD6-AP bound to cells expressing CELSR1 and CELSR2, but not to cells expressing CELSR3, consistent with the specificity seen in ACAPseq (*Figure 2I–K* and *Figure 4G*).

## Mode of homophilic binding by PCDH9

One PPI identified by ACAPseq – the homophilic binding of PCDH9 – was studied in depth. A structural analysis of the four amino-terminal cadherin domains of PCDH19, a non-clustered protocadherin, has demonstrated a distinctive mode of anti-parallel homophilic binding via an extended linear arrangement, with domains 1, 2, 3, and 4 of one monomer contacting domains 4, 3, 2, and 1, respectively, of the second monomer (*Cooper et al., 2016*). A nearly identical mode of dimerization is observed among clustered protocadherins (*Goodman et al., 2016a*; *Goodman et al., 2016b*; *Nicoludis et al., 2016*).

Preliminary binding assays with PCDH9-Fc and PCDH9-AP, using a 96-well plate assay with protein-G captured Fc, showed very weak signals, consistent with the relatively low affinities characteristic of cadherin and proto-cadherin homophilic binding (e.g. ~20 µM and ~160 µM for N- and E-cadherin, respectively; *Katsamba et al., 2009*). Using aggregation of PCDH9-Fc coated Protein-G magnetic beads, an assay with higher avidity, we observed robust homophilic binding with the full ECD and with deletion derivatives lacking cadherin domains 5, 6, and 7, either singly or in combination (*Figure 9A–C*; immunoblots with all of the proteins used in the experiments in *Figure 9* are shown in *Figure 9—figure supplement 1*, and semi-quantitative summaries of the bead binding assay results are in Table S3). Deletion of cadherin domains 1 or 4 dramatically reduced bead aggregation. In contrast, deletion of cadherin domains 2 or 3 led to only a modest decrement in bead aggregation.

To extend this analysis, we assessed interactions between pairs of PCDH9 ECD mutants, by coating magnetic beads with one PCDH9-Fc mutant and smaller red fluorescent microspheres with a second PCDH9-Fc mutant (*Figure 9D and E*). Using LYPD6-Fc as a negative control, we observed strong and specific aggregation if one set of beads was coated with a deletion of cadherin domain one and the second set of beads was coated with a deletion of cadherin domain 4. Substantial aggregation was also observed if one set of beads was coated with a deletion of cadherin domain two and the second set of beads was coated with a deletion of cadherin domain 3. In agreement with the magnetic bead aggregation in *Figure 9B and C*, the magnetic plus fluorescent bead aggregation assay showed greatly reduced aggregation when both sets of beads were coated with a mutant deleted for cadherin domain 1 or when both sets of beads were coated with a mutant deleted for cadherin domain 4.

Taken together, these data support an extended mode of anti-parallel homophilic binding for PCDH9 in which domains 1 – 4 are arranged in the same manner as for PCDH19 and the clustered protocadherins (*Figure 9F*). As summarized in *Figure 9G*, in the context of this model, derivatives lacking either domains 2 or 3 are able to form a reasonably stable anti-parallel complex, suggesting that interactions between domains 1 and 4 may contribute more significantly to binding affinity than the interactions between domains 2 and 3. Additionally, the interaction between three contiguous cadherin domains in their normal configuration (e.g. domains 2-3-4 in one protein with domains 1-2-3 in a second protein) produces a level of bead aggregation that is similar to the level observed with the full ECD.

## Discussion

### Advantages and limitations of ACAPseq

ACAPseq has a number of favorable attributes. First, ACAPseq interrogates the entire repertoire of translated proteins from the tissue of interest, including all splice isoforms, in a single reaction. Second, ACAPseq is extremely sensitive. Starting with 40 ug of polyribosome input, the starting abundance of target mRNAs can be as low as one part in 100,000 (~10 pg mRNA; for example Celsr1 and Flt1). For such rare mRNAs, the amount of specific mRNA captured is ~1 pg. Third, even a modest enrichment (e.g. 20–40 fold) of RNAseq reads in the bait of interest vs. control baits provides a signal that is substantially above background. Fourth, polyribosome capture depends on binding but not on biological function. Fifth, the high valency of the bait-coated magnetic beads and the polyvalent nature of the polyribosome suggest that relatively weak interactions can be detected. Sixth,

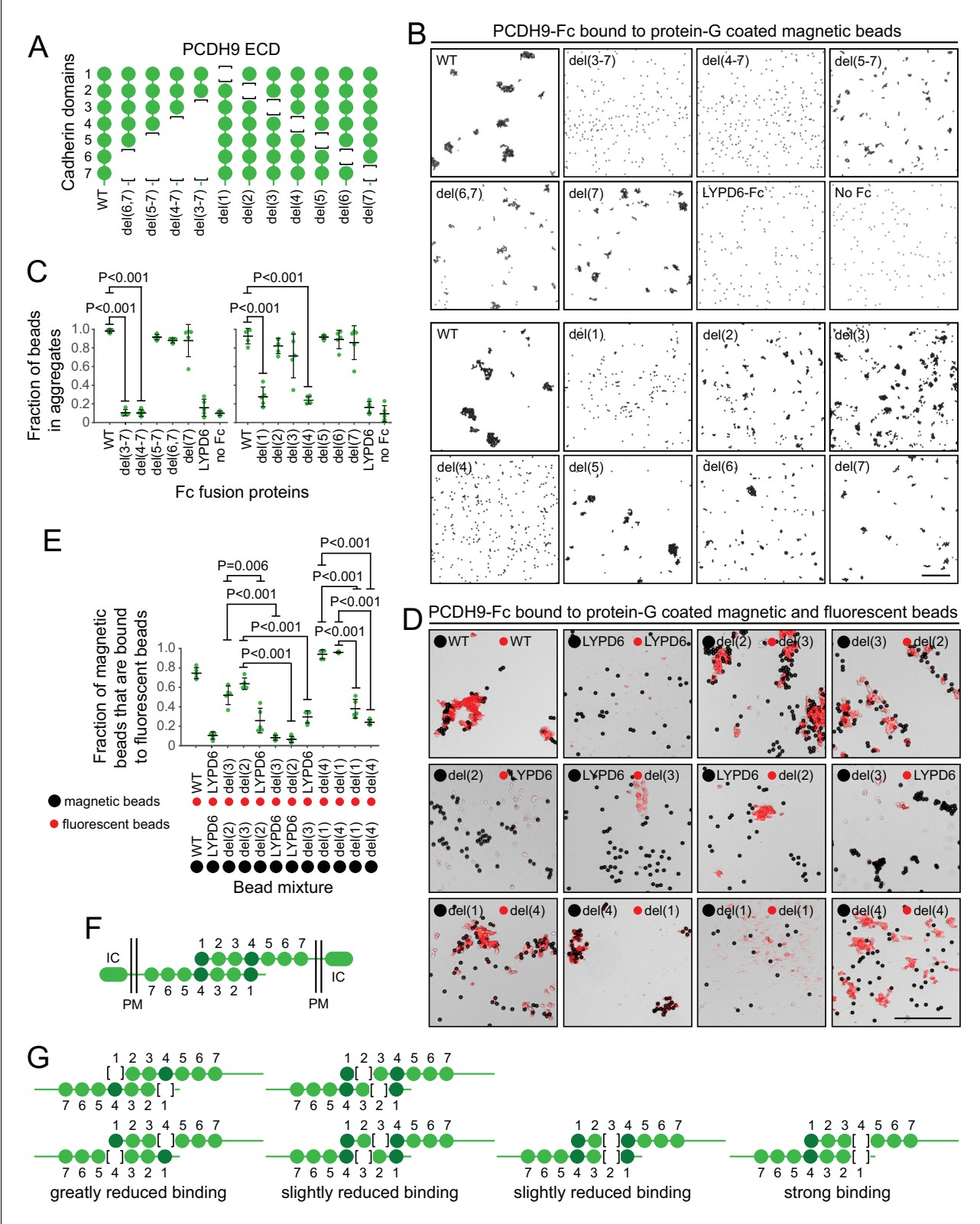

**Figure 9.** Molecular basis of PCDH9 homophilic binding. (A) Deletion derivatives of the ECD of PCDH9-Fc. Immunoblots of WT and mutant proteins are shown in *Figure 9—figure supplement 1*. (B) Aggregation of Protein-G-magnetic beads coated with the indicated PCDH9-Fc fusions. Control beads were either coated with LYPD6-Fc or were uncoated. Scale bar, 50 um. (C) Quantification of the fraction of beads that were aggregated (i.e., two or more beads in direct contact) from five representative images per condition. For (C) and (E), each data point represents 200 – 300 beads from one

*Figure 9 continued on next page*

*Figure 9 continued*

image; bars, mean and standard deviation; and N = 5 for each group. (**D**) Aggregation of Protein-G-magnetic beads (large black spheres) and red fluorescent microspheres (small red spheres) coated with the indicated PCDH9-Fc fusions. Control beads were coated with LYPD6-Fc. Bead aggregates spread out when placed on a slide and cover-slipped, distorting the geometry of bead contacts. Scale bar, 50 um. (**E**) Quantification of the fraction of magnetic beads that were in contact with one or more red fluorescent beads from five representative images per experiment. (**F**) Model for the anti-parallel homophilic binding between PCDH9 domains 1 – 4. (**G**) Summary of bead aggregation data with single domain mutants. Bead binding experiments were performed independently three times and were highly reproducible. Images and quantification are from one set of experiments.

DOI: https://doi.org/10.7554/eLife.40982.017

The following figure supplement is available for figure 9:

**Figure supplement 1.** Immunoblots of secreted PCDH9-Fc ECD domain fusions used for bead binding assays in *Figure 9*.

DOI: https://doi.org/10.7554/eLife.40982.018

ACAPseq surveys protein products representing the full range of mRNA transcript diversity in the starting polyribosome population. An analogous survey with high throughput binding to immobilized fusion proteins would require a plasmid collection coding for the full set of protein isoforms for all potential baits. ACAPseq could also be used to explore the effects of naturally occurring genetic diversity on target binding by starting with a genetically heterogenous pool of polyribosomes, for example, by mixing polyribosomes from diverse strains or from different species.

In its current form, ACAPseq also has limitations, as indicated by (1) the presence of multiple targets that pass the TPM thresholds but are likely to be false positives and (2) the failure of multiple baits to identify known PPIs, that is, false negatives. These are discussed in the paragraphs that follow.

False positives might have a variety of explanations. First, false positives could reflect interactions with unfolded or partially folded polypeptide targets that might normally be resolved in vivo by chaperones or by interactions with bone fide binding partners but that persist in vitro under ACAPseq binding conditions. Second, false positives could reflect adventitious interactions between proteins that are spatially sequestered in vivo, for example between a secreted protein bait and a cytoplasmic protein target. Third, false positives could arise from a non-optimal setting and/or weighting of threshold parameters. As illustrated by the analysis of sensitivity, specificity, and FDR for VEGF and EFNA1 baits, the number of false positives is highly sensitive to these parameters. The wide range of affinities underlying different PPIs imply that it would be useful to explore different TPM cut-offs or weightings for different baits.

False negatives might also have a variety of explanations. First, some of the baits may not have a polypeptide binding partner – that is, the negative finding is a true negative. Second, the binding partner might not be expressed at high enough levels (or at all) for its mRNA to be detected. In the present study, this explanation seems particularly plausible for those baits that are not detectably expressed in the mouse brain, including FGF21, HEPACAM2, TIGIT, VSIG1, and VSIG8. Third, the configuration of the bait-Fc fusion protein may not allow bait-target binding, as seen in *Figure 6—figure supplement 1* for VEGF-Fc and NRP1. Fourth, the critical binding domain(s) in the target protein might not be correctly folded in the context of the nascent polypeptide. Studies of several artificial and natural multi-domain proteins have revealed native folding of individual domains while the nascent polypeptides are still associated with the ribosome (*Chen et al., 1995*; *Han et al., 2012*; *Holtkamp et al., 2015*), but the extent to which these observations generalize to the rest of the proteome is not known. Fifth, binding could require co-assembly of two or more polypeptides or could require post-translational modifications that do not occur (or do not occur efficiently) in the context of the nascent polypeptide. For example, post-translational modifications that occur in the Golgi apparatus would be absent from polyribosome-bound nascent polypeptides, which are derived from the rough endoplasmic reticulum. Sixth, a PPI could occur with a domain of the target protein that is too close to the carboxy-terminus for efficient binding when the nascent polypeptide is bound to the ribosome. Conveniently, many cell-surface proteins are oriented with their amino-termini facing the extracellular space (type one topology; for example tyrosine kinase receptors), and this is the optimal arrangement for capturing their nascent polypeptides with ECD baits, as used here. Seventh, the ACAPseq signal may be too small to rise above the noise of non-specific interactions. This last point can be appreciated by assessing the signal-to-noise ratio in the ASPs in *Figure 4*, and it suggests that modifications to the protocol to reduce non-specific sticking and enhance

reproducibility of polyribosome capture could reveal both lower affinity interactions and lower abundance mRNAs.

## Prospects for improvements and future applications

The possibility that bait binding to nascent polypeptides might be limited by protein folding efficiency or post-translational modifications suggests that the addition of chaperones or modifying enzymes to the polyribosome sample might improve ACAPseq performance. Alternately, if unfolded or incompletely folded domains in nascent polypeptides are bound to chaperones and if the chaperone-polypeptide complex occludes access by the bait to adjacent folded domains, then removing chaperones from the polyribosomes might improve ACAPseq performance.

It will be of interest to determine how well ACAPseq performs with intracellular proteins as baits and with baits that consist of molecules or structures other than single proteins. For example, baits could consist of naturally occurring or synthetic organic molecules or polymers (e.g. carbohydrates), or complex biological structures such as intact viruses, bacteria, or fungi.

For humans and a small number of model organisms, extensive collections of sequenced cDNA clones currently exist and can be used for biochemical PPI screens, as seen, for example, in *Hsu et al. (2017)*. However, for the vast majority of organisms, including most microbes, plants, and insects, such collections do not exist. ACAPseq may be especially suitable for PPI discovery in these organisms because the nascent polypeptides displayed on polyribosomes present a target pool that approximates the complexity of the entire proteome.

In sum, ACAPseq is a novel PPI discovery platform that could complement existing PPI discovery platforms.

## Materials and methods

### Mice

All mice were housed and handled according to the approved Institutional Animal Care and Use Committee (IACUC) protocol MO16M367 of the Johns Hopkins Medical Institutions.

### Plasmids

Bait-Fc plasmids were prepared by cloning the PCR amplified ECD region of the protein of interest into a CMV immediate early region enhancer/promoter-based expression plasmid (pRK5) with a human IgG1 Fc coding region distal to the insert. PCR templates consisted of either (1) full-length cDNA plasmids or (2) a mixture of first strand cDNA prepared from gestational day 15 embryos and postnatal day three mouse brains. Seven bait-Fc plasmids were a kind gift of Dr. Woj Wojtowicz (*Visser et al., 2015*). The four CNTN-Fc plasmids were a kind gift of Dr. Davide Comoletti (Rutgers University). The standard bait-Fc fusion construct consisted of sequences coding for the full amino-terminal extracellular domain including the endogenous signal peptide, a 22 amino acid linker derived from the region immediately distal to the cysteine-rich domain of mouse Frizzled8, and the Fc region of human IgG1 beginning with the hinge (*Aruffo et al., 1990*).

### Buffers, tissue, and plasticware

All reagents for ACAPseq reactions were RNAse free. Disposable plasticware was assumed to be RNAse-free. Incomplete homogenization buffer (IHB) is 25 mM Tris, pH = 7.5, 150 mM KCl, 15 mM MgCl$_2$, 250 mM sucrose, and was stored in aliquots at $-80°$C. To prepare complete homogenization buffer (CHB), the following was added on the day of the experiment: 0.1% vol of 100 mg/ml cyclohexamide, 1% vol 100 mg/ml heparin, 5% vol 200 mM ribonucleoside vanadyl complex (New England Biolabs), one tablet Roche complete protease inhibitor per 50 mls. The final concentrations in CHB are: 100 ug/ml cyclohexamide, 1 mg/ml heparin, 10 mM ribonucleoside vanadyl complex, and 1X protease inhibitors. Addition of ribonucleoside vanadyl complex is optional for tissues that have low ribonuclease levels, such as neonatal brain. Resuspension buffer (RB) is identical to CHB except that it lacks ribonucleoside vanadyl complex and sucrose, and it contains 0.5% Igepal-630% and 0.5% sodium deoxycholate. Binding buffer (BB) is identical to RB, except that it additionally contains 1 mM CaCl$_2$ and 250 mM sucrose.

Neonatal mouse brains were dissected, snap frozen in a tube in a dry ice/ethanol bath, and stored at −80℃. For polyribosome preparations, all solutions were ice cold, and samples were kept cold by placing the glass homogenizer and tubes on ice. Centrifuges, centrifuge rotors, and centrifuge buckets were pre-chilled before use. End-over-end rotations and micro-centrifugations were performed in the cold room. Other procedures (e.g., tissue homogenization) were performed outside of the cold room with chilled reagents. All buffers in contact with polyribosomes contained 100 ug/ml cyclohexamide and 1 mg/ml heparin (a general RNAse inhibitor).

## Polyribosome preparation

To prepare polyribosomes from 9 grams of frozen brain tissue, frozen tissue was powdered between two metal blocks chilled with liquid nitrogen or dry ice. This was done by striking the upper block with a hammer. The powdered tissue was homogenized in five volumes (45 mls) CHB, with eight strokes of a homogenizer fitted with a motor-driven Teflon pestle. The pestle was rotated at ~100 rpm during homogenization. The homogenate was centrifuged at 1000 x g in a swinging bucket rotor at 4℃ for 20 min. The supernatant was collected and the nuclear pellet discarded. One-tenth volume each of 10% Igepal-630% and 10% sodium deoxycholate (final concentration of each is 1%) was added to the supernatant and gently mixed at 4℃ for 10 min with end-over-end rotation. The detergent solubilized homogenate was centrifuged at 16,000 x g for 15 min at 4℃, and the supernatant was collected, care being taken to avoid the residual nuclei in the pellet.

~30 mls of the supernatant was gently loaded onto a two-part sucrose shelf (3 ml of 2.5 M sucrose underneath 1.5 ml of 2.0 M sucrose, each in 25 mM Tris, pH = 7.5, 150 mM KCl, 15 mM $MgCl_2$ with 100 ug/ml cyclohexamide, 1 mg/ml heparin, 1X protease inhibitors, and with 1% each of Igepal-630 and sodium deoxycholate added on the day of the experiment) in an SW28 tube. The location of the sucrose interfaces was marked on the side of each tube, and the tubes were topped off with CHB containing 1% each of Igepal-630 and sodium deoxycholate. Samples were centrifuged at 20,000 rpm, for 17 hr, at 4℃. At the end of the run, the polyribosomes are present in the 2.5 M sucrose shelf, but are not pelleted to the bottom of the tube (*Kraus and Rosenberg, 1982*). We favor the use of a 2.5 M sucrose shelf over the standard polyribosome preparation procedures that produce a pellet of polyribosomes at the bottom the tube because, in our experience, ACAPseq is significantly more efficient if the polyribosomes have not been pelleted. After centrifugation, the supernatant layer was removed with a pipette, the walls of the tube were carefully washed with 3 ml resuspension buffer (RB), and the RB and the 2 M sucrose layer were then removed with a wide-bore pipette tip. Chromatin from residual nuclei in the sample resides in the upper sucrose layer and contributes to its viscosity. The walls of the tube were washed again with 3 ml RB, and then the RB was removed from the top of the 2.5 M sucrose shelf.

The 2.5 M sucrose layer was transferred to a 50 ml tube, 10 volumes of RB was added to lower the sucrose concentration to 0.25 M, and the sample was mixed well. A 100 ul aliquot was removed, clarified by microcentrifugation at 10,000 rpm for 5 min at 4℃, and 25 ul of the clarified sample used for RNA extraction with the RNAeasy Plus Micro kit (Qiagen) for quality control (gel electrophoresis and optical density) and NextGen sequencing. The remaining ~25 ml of polyribosomes was divided into 1 ml aliquots, snap frozen in a dry-ice ethanol bath, and stored at −80℃.

## Preparation of bait-Fc proteins

Bait-Fc proteins were produced in serum-free DMEM/F12 supplemented with Hepes, sodium bicarbonate, and penicillin/streptomycin (SFM) when using adherent HEK293T cells, or in Freestyle medium (Thermo-Fisher/Gibco 12338) with penicillin/streptomycin when using suspension cells (HEK293F). Bait-Fc protein production was first tested on a small scale and the yield estimated based on either (1) capturing the bait-Fc from 1 ml of conditioned SFM (SFCM) using Protein-G sepharose beads and analyzing the captured protein on SDS-PAGE followed by Coomassie Blue staining, or (2) running 10 – 20 ul of SFCM on SDS-PAGE followed by immunoblotting with anti-human Fc. Bait-Fc proteins with yields greater than ~0.5 ug/ml were used for ACAPseq.

For larger scale protein production, 2 – 4 10 cm plates with adherent HEK293T cells or 50 – 100 mls of a suspension culture of HEK293F cells were transiently transfected with the bait-Fc expression plasmid. For adherent HEK293T, cells were transfected at ~80% confluence using polyethyleneimine (PEI). One day after transfection, each plate was washed 3 times with 5 ml of SFM to remove calf

serum, before adding 5 ml SFM per plate. Extensive washing is essential to reduce the concentration of bovine IgG, which is ~10 mg/ml in serum and competes for binding to Protein-G magnetic beads. At daily intervals for the next 2 – 3 days, SFCM was collected and replaced with 5 ml fresh SFM per plate. SFCM was stored at 4°C for up to several months after filtering through a 0.2 um filter and adding sodium azide to 0.01%, or at −80°C for longer intervals. The bait-Fc concentration was estimated as described in the preceding paragraph.

### Capturing polyribosomes on Protein-G magnetic beads.

Approximately 10 ug of each bait-Fc in SFCM was used to coat Protein-G magnetic beads (75 ul of 30 mg/ml Protein-G Dynabeads; Thermo-Fisher). The Dynabeads were first washed twice with PBS or SFM. For each bait-Fc, washed Dynabeads were incubated with 5 – 20 ml of SFCM overnight at 4°C with end-over-end rotation. The following day, the bait-Fc-Dynabeads were washed once with PBS, transferred to a 1.5 ml microfuge tube, and washed once with 1 ml binding buffer (BB).

Polyribosome capture was generally performed with 8 – 10 baits in parallel, with EFNA1-Fc as one of the baits in each group. Bait-Fc-coated Dynabeads were incubated with polyribosomes (40 ug RNA, determined by $OD_{260}$) in 600 ul BB in a 1.5 ml tube. The tube tops were secured with parafilm and the tubes were rotated end-over-end overnight at 4°C. The next day, the Dynabeads were washed 4 times with BB, and the bound RNA was extracted with an RNeasy Plus Micro Kit. RNA was eluted twice with 11 ul of water. Yields were ~5 ng per sample. Bar-coded libraries for NextGen sequencing (Illumina) were prepared using the Ovation Single Cell RNAseq Library Kit (NuGen). Sequencing was performed with 50 bp single-end reads, and ~10 million reads per library.

A rough cost analysis for materials includes (1) cell culture plasticware and medium for producing 10 – 30 ug of bait protein, (2) materials for RNA purification, quality control, and small-scale library construction, (4) Illumina sequencing costs for 10 million reads. In 2017, this came to ~$400 per sample.

### Analysis of RNAseq data

RNAseq reads were aligned to the GRCm38/mm10 mouse genome using the RSEM-1.3.0 program with the command ~/RSEM-1.3.0/rsem-calculate-expression -p 20 –bowtie2 –bowtie2-path ~/bowtie2-2.2.9 –estimate-rspd –append-names –output-genome-bam –sort-bam-by-coordinate (*Li and Dewey, 2011*; *Ghosh and Chan, 2016*). Bigwig files for visualization on the IGV browser were produced using the deepTool program bamCoverage (*Ramírez et al., 2016*), with the command bamCoverage -p 20 -bs 1 —normalizeUsingRPKM. For the scatter plot analyses (e.g., *Figure 4*), only those reads mapping to the exons of protein-coding transcripts were included. The plots appeared nearly identical with inputs of total exonic read counts, reads per kilobase million (RPKM), or transcripts per million (TPM) (https://statquest.org/2015/07/09/rpkm-fpkm-and-tpm-clearly-explained/). Therefore, TPM was used throughout. For comparison of transcript abundance across two samples, statistical significance was calculated using RSEM-1.3.0 and EBseq-1.2.0 program following the suggested instructions (*Leng et al., 2013*). A false discovery rate (1-PPDE) smaller then 0.05 was considered significant. PPDE, posterior probability deferentially expressed. RNAseq data has been deposited in the Gene Expression Omnibus (GEO) at NCBI (Accession number GSE121524).

### Binding assays with alkaline phosphatase fusion proteins: live cell binding

The standard AP-fusion construct consists of sequences coding for an amino-terminal fusion partner with its endogenous signal peptide followed by a glycine/serine linker, a myc epitope tag, and the coding region of mature human placental alkaline phosphatase (PLVAP) without GPI-anchoring sequences at its carboxy-terminus (*Smallwood et al., 2007*). AP-fusion proteins were expressed and harvested in SFCM as described above for bait-Fc fusion proteins.

For cell binding, HEK293T cells were grown on gelatin-coated wells in 12 well plates and transiently transfected with plasmids coding for the target protein of interest and a tdTomato control. 48 hr after transfection, the medium was removed, and 0.6 ml of ice-cold SFCM containing the AP-fusion was added to each well. The plate was gently rotated horizontally for 2 hr at 4°C, then washed 5 times with PBS, fixed in PBS with 4% paraformaldehyde for 15 min at 4°C, and washed twice with PBS. After covering the inner surface of the plate's lid with parafilm and wrapping the plate in foil,

the plate was floated in a 70°C water bath for 1 – 1.5 hr to heat inactivate endogenous phosphatases. After heat inactivation, the PBS was replaced with 0.5 ml AP buffer [0.1 M Tris, pH 9.5, 50 mM MgCl$_2$, 0.1 M NaCl] containing 1.65 μl 4-nitro blue tetrazolium (Roche 11383213001), and 1.65 μl 5-bromo-4-chloro-3-indolyl (Roche 11383221001) per well. The plate was covered with aluminum foil and gently rotated horizontally at room temperature. The AP reactions were stopped after several hours.

### Bead aggregation assays: homophilic binding with magnetic beads.

To test homophilic binding of WT or deletion variants of PCDH9-Fc, 0.4 μl of 30 mg/ml PBS-washed Protein-G coated Dynabeads (Invitrogen 10004D) was added to 100 μl SFCM in a 1.5 ml microfuge tube, and rotated about the long axis of the tube with that axis maintained at a 20 degree angle to the horizontal plane for 1.5 to 2 hr at room temperature. 5 – 10 μl of the solution was pipetted from the bottom of the tube, placed on a glass slide, cover-slipped, and examined with a Zeiss SM700 confocal microscope. Images were exported to ImageJ to manually quantify bead aggregation.

### Bead aggregation assays: heterophilic binding with magnetic and fluorescent beads

To test the binding between various pairs of deletion variants of the PCDH9-Fc fusion, 0.8 μl of 30 mg/ml PBS-washed Protein-G coated Dynabeads was added to 100 μl SFCM containing the first PCDH9-Fc fusion protein and 10 ul of PBS-washed Protein-G Coated Florescent Nile Red microspheres (Spherotech PGFP-0556 – 5; 0.54 um diameter; 0.1% weight/volume stock solution) was added to 100 μl SFCM containing the second PCDH9-Fc fusion protein. The tubes were rotated as described in the preceding paragraph for 1.5 to 2 hr at room temperature. The beads were then washed twice in SFM, resuspended in 50 μl SFM, mixed together, and rotated for an additional 1.5 to 2 hr at room temperature. 5 – 10 μl of the bead mixture was examined and analyzed as described above for the magnetic bead aggregation assay.

### Affinity capture of TNR

48 hr after transient transfection of HEK293T cells, cells were washed with PBS, lysed in RIPA buffer [10 mM Tris, pH = 8.0, 1 mM EDTA, 1% TritonX-100, 0.1% sodium deoxycholate, 0.1% SDS, 140 mM NaCl, 1 mM PMSF, and 1X Protease inhibitors (Roche)] with 0.1% bovine serum albumin (BSA) and 1x protease inhibitor cocktail (Roche) by gentle pipetting, and rotated end-over-end for one hour at 4°C. The lysate was then clarified by centrifugation at 10,000 x g for 10 min, and an aliquot (corresponding to 100 ug of cellular protein) used per capture reaction. 80 ul protein-A sepharose beads were washed twice with PBS, incubated with 1 ml of LRTM2-Fc SFCM with end-over-end rotating for three hours at 4°C, and washed twice with RIPA buffer with 0.1% BSA. The LRTM2-Fc-coated protein-A sepharose beads ('beads' hereafter) were evenly divided among the four aliquots of clarified cell lysates, 1 ml of RIPA buffer with 0.1% BSA, 1 mM CaCl$_2$, and 1 mM MgCl$_2$ was added to each aliquot, and the samples were rotated overnight at 4°C. The next day, the beads were washed three times with RIPA buffer with 0.1% BSA, 1 mM CaCl$_2$, and 1 mM MgCl$_2$, and then twice with RIPA buffer with 1 mM CaCl$_2$ and 1 mM MgCl$_2$. 35 ul of 2x SDS sample buffer was added to each tube and the tubes heated to 100°C for two minutes, and briefly centrifuged to pellet the beads. The released proteins were resolved by SDS-PAGE and immunoblotted for the OLLAS epitope (*Park et al., 2008*).

### Statistical analysis

P-values were calculated with a 2-tailed student's t-test. Log$_{10}$ plots are calculated with TPM +1 rather than TPM values so that any TPM values of zero are represented by a finite log$_{10}$ value. IBM SPSS Statistics 24 was used for statistical analysis.

## Acknowledgements

The authors are grateful to Chris Cho and James Berger for assistance with protein structure visualization; Solomon Snyder for sharing his 96-well plate reader; Danelle Devenport, Elaine Fuchs, and Tadashi Uemura for CELSR cDNAs; Davide Comoletti for CNTN cDNAs; Woj Wojtowicz for sharing

her collection of Fc fusion plasmids; the JHMI Deep sequencing core; Elena Pasquale for advice; and Allen Buskirk, Daniel Goldman, and Boris Zinshteyn for helpful comments on the manuscript.

Supported by the Arnold and Mabel Beckman Foundation, NEI/NIH (R01EY018637), the David Labovitz Fund, the Chinese Scholarship Council, and the Howard Hughes Medical Institute.

## Additional information

### Competing interests
Jeremy Nathans: Reviewing editor, *eLife*. The other authors declare that no competing interests exist.

### Funding

| Funder | Grant reference number | Author |
| --- | --- | --- |
| Howard Hughes Medical Institute | | Jeremy Nathans |
| National Eye Institute | R01EY018637 | Jeremy Nathans |
| Arnold and Mabel Beckman Foundation | | Jeremy Nathans |
| David Labovitz Fund | | Jeremy Nathans |
| Chinese Scholarship Council | | Jeremy Nathans |

The funders had no role in study design, data collection and interpretation, or the decision to submit the work for publication.

### Author contributions
Xi Peng, Data curation, Formal analysis, Investigation, Writing—review and editing; Francesco Emiliani, Investigation, Methodology; Philip M Smallwood, Hong Lei, Mark F Sabbagh, Investigation; Amir Rattner, Data curation, Formal analysis, Supervision, Investigation, Visualization, Methodology, Writing—review and editing; Jeremy Nathans, Conceptualization, Formal analysis, Supervision, Funding acquisition, Investigation, Methodology, Writing—original draft, Project administration, Writing—review and editing

### Author ORCIDs
Mark F Sabbagh (iD) http://orcid.org/0000-0003-1996-5251
Jeremy Nathans (iD) http://orcid.org/0000-0001-8106-5460

### Ethics
Animal experimentation: This study was performed in strict accordance with the recommendations in the Guide for the Care and Use of Laboratory Animals of the National institutes of Health. All of the animals were handled according to the approved Institutional Animal Care and Use Committee (IACUC) protocol MO16M367 of the Johns Hopkins Medical Institutions.

### Decision letter and Author response
Decision letter https://doi.org/10.7554/eLife.40982.027
Author response https://doi.org/10.7554/eLife.40982.028

## Additional files

### Supplementary files
• Supplementary file 1. Baits and captured targets. DC, plasmids from Davide Comoletti. WW, plasmids from Woj Wojtowicz. The threshold for target interaction is: (1)>35 captured TPMs, (2) captured TPMs > 3 x the maximal number of TPMs captured with any of the other baits used in the

same experimental cohort, (3) captured TPMs > 3 x the TPMs in the unselected polyribosomes, and (4) the target is a secreted or membrane protein (as designated by https://www.proteinatlas.org/humanproteome/secretome). The following baits used the mouse Fz8 signal peptide: ASTM1, ASTN2, EPO, FGF19, FGF21, VEGF-Fc. All other baits used the endogenous signal peptide. The last six amino acids of the bait sequence is shown for those baits that are derived from a protein with an amino-terminal extracellular domain followed by a trans-membrane domain. The following baits are human sequences: EPO, FGF19, FGF21, VEGF-Fc, Fc-VEGF. All others are mouse sequences.

DOI: https://doi.org/10.7554/eLife.40982.019

• Supplementary file 2. ACAPseq with six baits: abundance and fold enrichment of targets mRNAs (none)

DOI: https://doi.org/10.7554/eLife.40982.020

• Supplementary file 3. Binding properties of PCDH9 mutants. The PCDH9 binding data refer to *Figure 9*.

DOI: https://doi.org/10.7554/eLife.40982.021

• Supplementary file 4. All supra-threshold targets for the 92 baits.

DOI: https://doi.org/10.7554/eLife.40982.022

• Transparent reporting form

DOI: https://doi.org/10.7554/eLife.40982.023

### Data availability

RNAseq data has been deposited in the Gene Expression Omnibus (GEO) at NCBI (Accession number GSE121524).

The following dataset was generated:

| Author(s) | Year | Dataset title | Dataset URL | Database, license, and accessibility information |
|---|---|---|---|---|
| Xi Peng, Francesco Emiliani, Philip M Smallwood, Amir Rattner, Hong Lei, Mark F Sabbagh, Jeremy Nathans | 2018 | Affinity capture of polyribosomes followed by RNAseq (ACAPseq), a discovery platform for protein-protein interactions | https://www.ncbi.nlm.nih.gov/geo/query/acc.cgi?acc=GSE121524 | Publicly available at the NCBI Gene Expression Omnibus (accession no. GSE121524) |

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
