## [Decision Letter]

Thank you for submitting your article "Affinity capture of polyribosomes followed by RNAseq (ACAPseq), a discovery platform for protein-protein interactions" for consideration by *eLife*. Your article has been reviewed by three peer reviewers, and the evaluation has been overseen by a Reviewing Editor and Philip Cole as the Senior Editor. The following individuals involved in review of your submission have agreed to reveal their identity: Kai Zinn (Reviewer #2); Rachelle Gaudet (Reviewer #3).

The reviewers have discussed the reviews with one another and the Reviewing Editor has drafted this decision to help you prepare a revised submission.

Summary:

This manuscript by Peng and colleagues provides a thorough evaluation of an RNAseq method, which they name ACAPseq, to identify protein-protein interactions using a large panel of extracellular domains of proteins as a test set. The methodology combines pulldown of polyribosomes from frozen tissue using Fc-tagged proteins as bait and the highly-multiplexed mRNA sequencing of the pulldown. Overall the experiments are convincing and the analyses appropriate. As described in the Discussion, this method has both strengths and weaknesses, but when implemented carefully and with understanding of the system's limitations it can provide useful information about potential protein-protein interactions of a protein of interest. Because many of the well-established techniques for discovery of protein-protein interactions do not work well for extracellular proteins, ACAPseq can fill a void in the protein-protein interaction discovery space. The method also has some clear advantages absent from other competing technologies: for example, the ability to link binding to specific splice variants.

Essential revisions:

We have the following suggestions to improve the manuscript.

1) 1209 candidate interactors seems like a lot to examine; but the authors might wish to point out that number is as high as it is mostly because a few baits identify many partners. It is likely that most of the interactions with baits that identify >10 partners are artifacts and indicate that the baits are "sticky." If you add up all the interactions for baits that identify <10 partners, there are only 187 of those, which is a more manageable number.

2) The authors might wish to indicate how much the sequencing cost. This looks like a lot of lanes, but I couldn't estimate how many samples were done. The cost of this method might be beyond the resources available to most labs. Related to this, can the authors provide an indication of the throughput? What was the size grouping for sequencing? Did they always sequence a pair (test bait plus EFNA-1 control)? Any recommendations based on their experience?

3) Two reviewers noted that Figures 9-11 don't really belong in this paper, as they are investigations of the binding sites on some of the candidates. These are probably included because they didn't have data for full papers on APP-CNTN3 (for example) and the other interactions and wanted to have a format in which to publish them. These figures make the paper rather long and unwieldy, and distract a bit from the description of the method. However the reviewers did not insist that they be removed. Perhaps the APP-CNTN3 section, which is particularly long, could be deleted or shortened.

4) Subsection “ACAPseq with diverse mammalian ECD baits”, second paragraph:

Supplementary file 1 should include additional columns for (1) database entry code for the relevant cDNA sequence and (2) the corresponding start and end positions of the cDNA (or corresponding translated protein) used in the bait construct.

5) "… in which domain 2 pairs with itself (in the domain 3 deletion mutant), domain 3 pairs with itself (in the domain 2 deletion mutant)…” etc.: these are unwarranted inferences, because the authors do not know which domain interacts with which domain in their constructs, only that the construct is or isn't sufficient to cause bead aggregation. This paragraph should thus be carefully reworded. A more conservative conclusion: it seems that the data in Figure 11 suggest that construct pairs in which a domain 1 – domain 4 interaction is possible have the strongest binding?

---

## [Author Response]

Essential revisions:We have the following suggestions to improve the manuscript.1) 1209 candidate interactors seems like a lot to examine; but the authors might wish to point out that number is as high as it is mostly because a few baits identify many partners. It is likely that most of the interactions with baits that identify >10 partners are artifacts and indicate that the baits are "sticky." If you add up all the interactions for baits that identify <10 partners, there are only 187 of those, which is a more manageable number.

Good points. The signal-to-noise challenge is non-trivial, and using a single set of threshold parameters for the entire set of 92 baits, as we did here, is useful for illustrative purposes but is not optimized for the different PPI affinities. A more in-depth analysis with different threshold parameters, as demonstrated for VEGF and EFNA1 in Figure 5, is needed to obtain the best signal-to-noise ratio for each bait. For example, using the threshold parameters defined in Supplementary file 1, EFNA1 had 21 candidate PPIs, 7 of which are bona fide partners. Similarly, LPHN1 had 21 candidate PPIs, 5 of which are bona fide partners. Thus, a cut-off of <10 partners, as suggested by the reviewer, would have eliminated these PPIs. We have expanded this section of the text to clarify these issues.

2) The authors might wish to indicate how much the sequencing cost. This looks like a lot of lanes, but I couldn't estimate how many samples were done. The cost of this method might be beyond the resources available to most labs. Related to this, can the authors provide an indication of the throughput? What was the size grouping for sequencing? Did they always sequence a pair (test bait plus EFNA-1 control)? Any recommendations based on their experience?

Those are good points. Costs include cell culture materials for production of 10-30 ug of bait protein and protein-G magnetic beads, which total ~$100 per bait. RNA purification and library preparation reagents cost ~$200 per sample. We sequence ~10 million reads per sample. 150 million reads on an Illumina NextSeq500 costs $1400, and, therefore, ~10 million reads costs ~$100. That adds up to ~$400 per bait for all material costs. We typically capture polyribosomes with 8-10 baits in parallel, and we always include a control EFNA1-Fc sample in the group. We have now added this information to the Materials and methods section.

3) Two reviewers noted that Figures 9-11 don't really belong in this paper, as they are investigations of the binding sites on some of the candidates. These are probably included because they didn't have data for full papers on APP-CNTN3 (for example) and the other interactions and wanted to have a format in which to publish them. These figures make the paper rather long and unwieldy, and distract a bit from the description of the method. However the reviewers did not insist that they be removed. Perhaps the APP-CNTN3 section, which is particularly long, could be deleted or shortened.

That is a fair critique. We have now removed the APP-CNTN3 experiments from the manuscript (text, 2 figures, and 1.5 supplemental figures). The APP-CNTN3 site-directed mutagenesis experiments are less central to the overall body of work. Additionally, APP-CNTN3 is a PPI that has been previously reported. We will publish it separately. However, we would like to keep the PCDH9 experiments in the paper as this is a previously unreported PPI, and the follow-up experiments are a nice validation showing that low affinity PPIs can be detected by ACAPseq.

4) Subsection “ACAPseq with diverse mammalian ECD baits”, second paragraph:Supplementary file 1 should include additional columns for (1) database entry code for the relevant cDNA sequence and (2) The corresponding start and end positions of the cDNA (or corresponding translated protein) used in the bait construct.

Thank you. We have added that information to the revised Supplementary file 1.

5) "… in which domain 2 pairs with itself (in the domain 3 deletion mutant), domain 3 pairs with itself (in the domain 2 deletion mutant)…” etc.: these are unwarranted inferences, because the authors do not know which domain interacts with which domain in their constructs, only that the construct is or isn't sufficient to cause bead aggregation. This paragraph should thus be carefully reworded. A more conservative conclusion: it seems that the data in Figure 11 suggest that construct pairs in which a domain 1 – domain 4 interaction is possible have the strongest binding?

Good points. We have reworked the text to describe the inferences more carefully, as indicated.